# Heat extremes in Western Europe increasing faster than simulated due to atmospheric circulation trends

Robert Vautard [1] ✉, Julien Cattiaux[2], Tamara Happé [3], Jitendra Singh[4], Rémy Bonnet [1], Christophe Cassou [5], Dim Coumou [1,3,6], Fabio D'Andrea[7], Davide Faranda [8], Erich Fischer [4], Aurélien Ribes [2], Sebastian Sippel [4] & Pascal Yiou [8]

Over the last 70 years, extreme heat has been increasing at a disproportionate rate in Western Europe, compared to climate model simulations. This mismatch is not well understood. Here, we show that a substantial fraction (0.8 °C [0.2°–1.4 °C] of 3.4 °C per global warming degree) of the heat extremes trend is induced by atmospheric circulation changes, through more frequent southerly flows over Western Europe. In the 170 available simulations from 32 different models that we analyzed, including 3 large model ensembles, none have a circulation-induced heat trend as large as observed. This can be due to underestimated circulation response to external forcing, or to a systematic underestimation of low-frequency variability, or both. The former implies that future projections are too conservative, the latter that we are left with deep uncertainty regarding the pace of future summer heat in Europe. This calls for caution when interpreting climate projections of heat extremes over Western Europe, in view of adaptation to heat waves.

Extreme heat has been increasing at global scale[1,2], with a rapid rate in several regions. In Western Europe[3], summer temperatures and heat extremes have warmed much faster than elsewhere in the mid-latitudes over the last two decades[3,4]. As a consequence, several unprecedented heatwaves took place in the last 20 years. In 2003, the full summer season mean temperature was unprecedented in Europe[5]. Northwestern Europe was hit by record temperatures in 2018[6,7]. In 2019, two short (3-day) but intense heat waves saw all-time temperature records broken in many places, associated with a rapid northward advection of Saharan air[6]. All-time records were broken again in 2022, with temperatures above 40 °C reaching far north (eg. Brittany, U.K.) (https://www.worldweatherattribution.org/without-human-caused-climate-change-temperatures-of-40c-in-the-uk-would-have-been-extremely-unlikely/, (2022)). Unprecedented, and even record-shattering extremes are plausible in climate projections[8], but the pace of their increasing magnitude in Western Europe is generally not predicted by these climate models, as well as trends in mean summer temperatures[4,9–12].

Here we focus on summer (JJA) maximum and mean of daily maximal temperatures (resp. denoted hereafter TXx and TXm for simplicity), and the regional amplification of their trends relative to the global temperature trend. Trends in TXx and TXm are calculated over the 73-year 1950–2022 period using a linear regression with the Global mean Surface Air Temperature (GSAT, see methods section) from

[1]Institut Pierre-Simon Laplace, CNRS, Université Paris-Saclay, Sorbonne Université, Paris, France. [2]Centre National de Recherches Météorologiques, Université de Toulouse, Météo-France, CNRS, Toulouse, France. [3]Institute for Environmental Studies, Vrije Universiteit Amsterdam, Amsterdam, Netherlands. [4]Institute for Atmospheric and Climate Science, ETH Zurich, Zürich, Switzerland. [5]Centre Européen de Recherche et de Formation Avancée en Calcul Scientifique, CNRS UMR 5318 Toulouse, France. [6]Royal Netherlands Meteorological Institute (KNMI), De Bilt, Netherlands. [7]Laboratoire de Météorologie Dynamique, IPSL, CNRS, Paris, France. [8]Laboratoire des Sciences du Climat et de l'Environnement, UMR 8212 CEA-CNRS-UVSQ, Université Paris-Saclay and IPSL, 91191 Gif-sur-Yvette, France. ✉e-mail: robert.vautard@ipsl.fr

ERA5, and are expressed in °C per global warming degree (GWD). As shown in Fig. 1 and Supplementary Fig. 1, both ERA5 reanalyses[13] and E-OBS interpolated observations[14] exhibit trends reaching more than 5 °C/GWD for TXx in northern France and Benelux. Over the limited area spanning 5W-15E; 45N-55N (blue box, called hereafter "Western Europe"), the land area-average TXx trend is 3.4 °C/GWD for ERA5 and E-OBS [2.4–4.3 °C/GWD]. It exceeds the more moderate TXm trends by about 40% for ERA5 (2.4 °C/GWD [1.7–3.0 °C/GWD] and 30% for E-OBS (2.6 °C/GWD [1.9–3.3 °C/GWD]). These rapid warming trends are exceptional on a global scale: The 20° × 10° Western Europe region has the highest TXx (all year round) trend of all regions of the same size around the globe between 75°S and 75°N shifted by steps of 5° (including sea points).

A variety of processes have been proposed for explaining these overproportional warming trends with respect to global temperature change. For mean summer temperatures, changes in mean atmospheric circulation[15,16], changes in aerosol[17] and changes in early summer soil moisture[18] and related feedbacks were considered for explaining (part of) the trends. For extreme heat, the increase in the frequency and persistence of split midlatitude jet states over the last 40 years, possibly associated with the reported weakening of the mean summer zonal circulation[19], can explain about a third of the amplified trend in heatwave intensity[3]. Changes in atmospheric circulations around Europe that favor heat were also emphasized[20,21], in particular a positive trend in a dipole structure with a low pressure over the Eastern Atlantic[22,23] and a high pressure over the Mediterranean extended towards central Europe[24]. Yet, no increasing trend was found in blocking over Scandinavia that has led to the 2018 heat wave[6,25]. Moreover, reported changes in Rossby waves are not robust and are sensitive to their exact definition[26]. In addition, variability of summer temperatures has been shown to be large in Central Europe[27]. Thus, while several studies have hinted at a potential role of dynamical

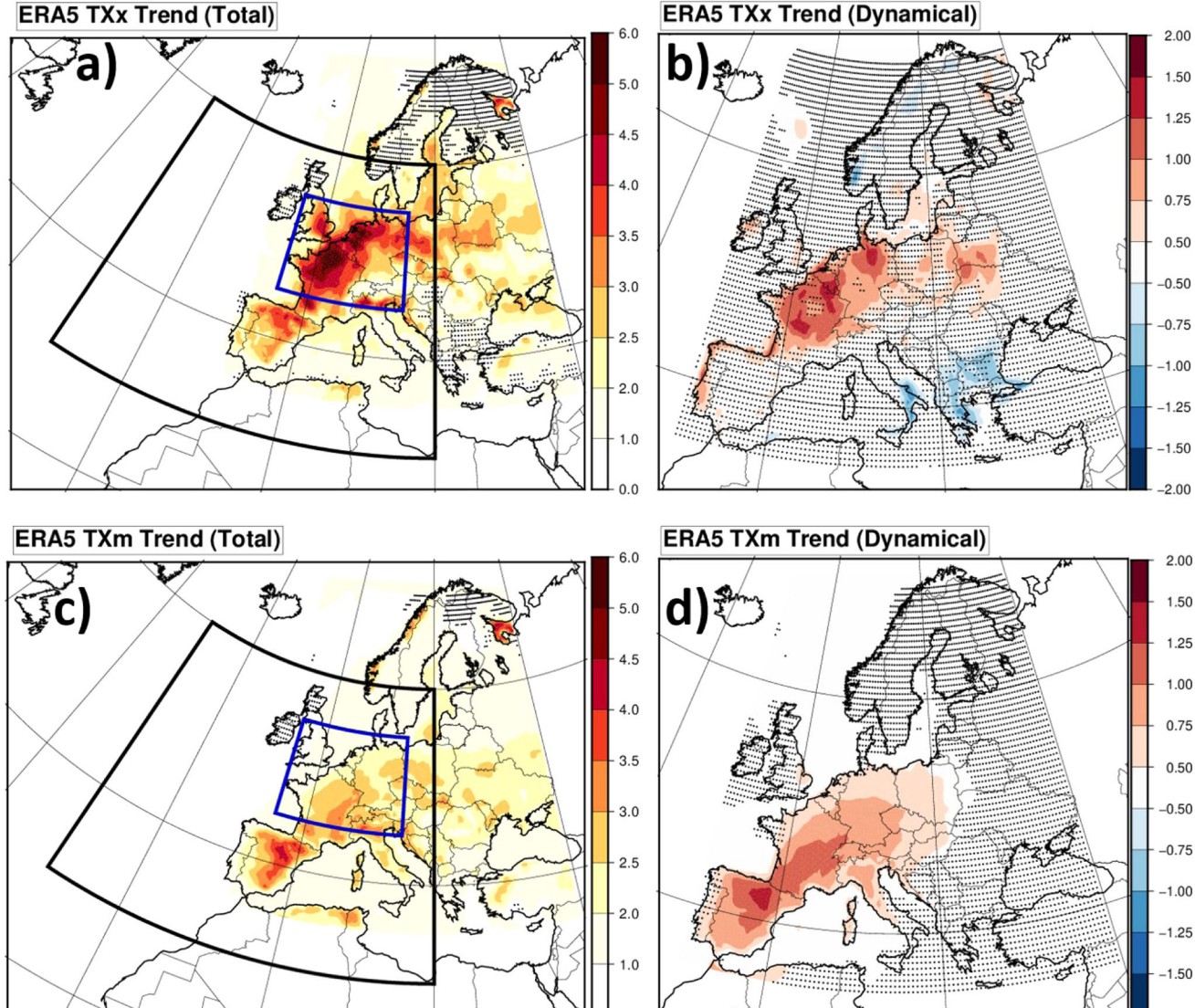

**Fig. 1 | Total and dynamical contributions to extreme and mean TX trends.** ERA5 reanalysis temperature trends relative to the global warming level (°C/GWD), for summer Maximum of maximal daily temperature (TXx) (**a**) and **b**) and summer Mean of maximal daily temperature (TXm, **c**) and **d**). The raw trend (**a**) and **c**) is compared to the estimated dynamical contribution to these trends (**b**) and **d**), obtained by replacing daily temperatures by those of best circulation analogues with a thermodynamic correction (see Methods). The areas highlighted are: (black box) the area used to calculate the anomaly correlation of 500 hPa streamfunction for the definition of analogues; the Western Europe focus area (blue box), where maximal daily temperature trends are averaged in this study. Dotted points show areas where statistical significance of trends is less than 95% (two sided). The statistical test uses a 2-sigma rule for the regression coefficient, accounting for the total number of well-separated analogues (see Methods).

changes in amplifying European heat waves, a systematic analysis is lacking, including also how models simulate these changes.

## Results

### Role of dynamical changes in the temperature trends

We used a method based on circulation analogues to assess the role of dynamical changes in the TXx and TXm trends (see the methods section for a full description). Regional atmospheric circulation patterns are characterized by their 500 hPa streamfunction over the domain shown in Fig. 1a (black box). We identify circulation analogues for a given day by searching for other summer dates (JJA months) with similar anomaly structures, measured by the spatial anomaly correlation coefficient (ACC). A set of dates with circulation analogues allows us to calculate statistics conditionally to a given circulation[28–31], or to assess the role of dynamical changes in circulation-conditioned variables[32,33].

In order to estimate the contribution of dynamical changes to TXx and TXm trends (called hereafter the "dynamical TXx and TXm trends"), we replace each daily temperature field by the temperature field from a different day that had the best analogue circulation. In the absence of long-term trends in circulation, this is equivalent to shuffling the temperature time series while keeping the dynamics, thereby creating a trend-free "analogue temperature time series". In the presence of long-term circulation trends, the trend in the analogue temperature time series comes from the changes in circulations (e.g. an increase in circulations favorable to heat, or vice versa). Replacement by analogues should in principle remove thermodynamical effects from global warming. As global warming is not homogeneous across the time period, and to ensure analogue regional temperatures represent a given global warming level, we further apply a correction by scaling all analogue temperatures to a reference year for global warming (2022) (see Methods). We verified that results were similar in both cases (with and without scaling).

The dynamical TXx trend (Fig. 1b) is generally positive over Western Europe and reaches about 1.5 °C/GWD in several areas. The dynamical TXm trend is found to exceed 1 °C/GWD over Southwestern Europe (Fig. 1d). Over Western Europe, the average TXm and TXx dynamical trends are respectively 0.74 °C/GWD [0.26–1.21 °C/GWD] and 0.79 °C/GWD [0.24–1.35 °C/GWD]. For E-OBS the dynamical trends are 0.78 °C/GWD [0.27–1.29 °C/GWD] and 0.86 °C/GWD [0.29–1.43 °C/GWD] for TXm and TXx respectively.

We verify these findings on the dynamical contributions to extreme temperatures trends with a second method, called "dynamical adjustment"[34]: The method uses a spatial circulation field (here: z500 for consistency with previous studies) as a proxy in order to estimate the contribution of circulation to temperature variability. Here, we use ridge regression, a linear regression technique that regularizes the coefficients of the high-dimensional circulation predictors[35], and we subsequently evaluate the dynamical contribution of z500 to the Western Europe TXx trends and averaged results over Western Europe (see method details in the Methods section). Results are consistent with the analogue approach (Supplementary Fig. 2), although with a slightly weaker dynamical TXx trend of 0.56 °C/GWD.

To test the sensitivity of our results to the analogue domain, we performed sensitivity experiments by extending and reducing the domain by 10° longitude and 5° latitude (leaving about 2/3 or more of the domain common with the reference one). The dynamical trend is significant and within 0.5 °C/GWD and 0.9 °C/GWD, except when reducing the domain towards the North-Eastern part (20W-20E;35N-60N), (dynamical tendency reduced to 0.38 °C/GWD) a probable consequence of the key role of the upstream part of the pattern.

Further, we investigate the specific streamfunction patterns associated with summer maximum extreme temperatures over central France [1.5E;46.5 N]—i.e., a region where the TXx dynamical trend is large (see Fig. 1). We select the reference date (29/06/2019) for which the streamfunction pattern (Fig. 2a) has a maximal average ACC (0.59) with other streamfunction patterns occurring each year when maximal temperature (TXx) is reached at this grid point, so it is most representative of those "TXx days". We find that about 15% of the summer days in total have an ACC larger than 0.5 with the 29/06/2019 pattern, and that 53 out of 72 other TXx patterns also correlate by more than 0.5. For the sake of simplification, we will refer this class of patterns as

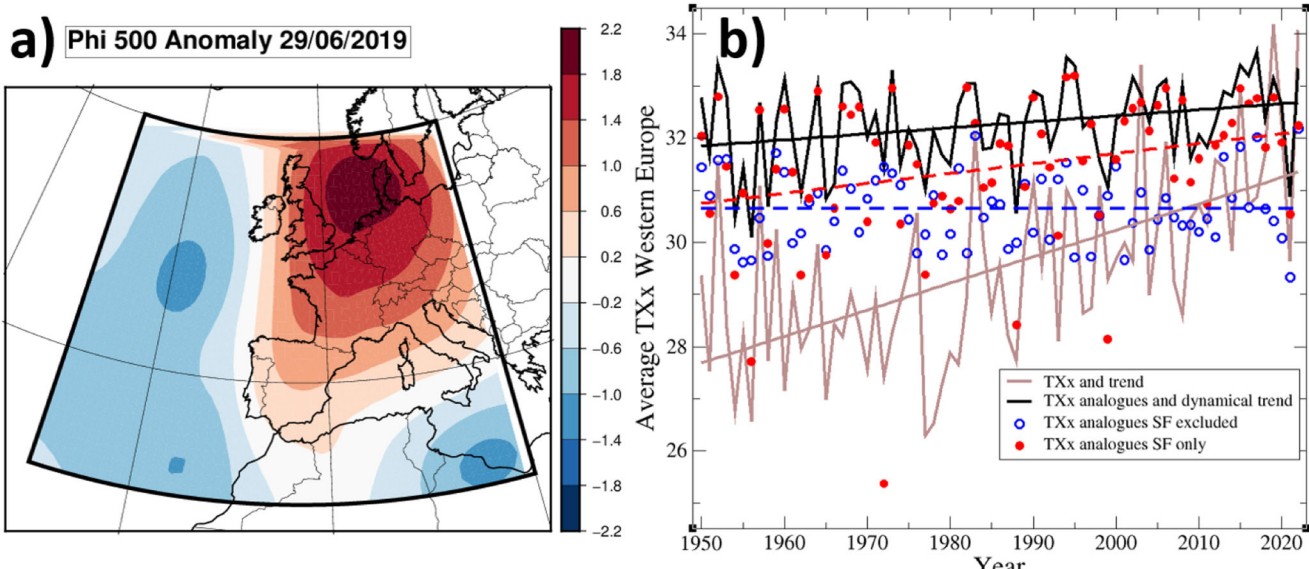

**Fig. 2 | Southerly flow anomalies and their contributions to summer temperature maxima. a** 500 hPa Streamfunction anomaly (Phi 500) of the 29/06/2019; **b** yearly time series of the Western Europe average of Summer maximal temperature TXx (brown), the TXx of the analogue time series, averaged over Western Europe and using the 3 best analogues (black curve) (see Methods), and the corresponding time series obtained by excluding (resp. including only) Southerly Flow

(SF) pattern dates before calculating the analogue TXx values (blue circles, resp. red circles). The sets of dates (SF dates or SF excluded dates) within a year over which the yearly maximum is sought are therefore complementary. In each case, analogues are calculated using the full set of patterns (i.e. for SF excluded dates, analogues may contain SF patterns). Linear trends for all series are also shown, with the same color as the series. The dashed trends are for SF-only or SF-excluded cases.

the "Southerly Flow" patterns (SF), since almost all of the patterns bear a positive west-east streamfunction gradient (eg. 99% of patterns when considering the gradient between 15°W and 5°E at 50°N), inducing southerly flows over the Western margin of Europe. This pattern also includes a strong anticyclonic component over Central Europe, which induces increased radiation and potential land-atmosphere feedbacks if persistent. As another example, the outstanding temperatures in London on 19/07/2022 were also obtained with a similar circulation pattern (ACC = 0.81 with 29/06/2019). To assess sensitivity to the reference pattern we also repeat all calculations with the 10 most representative TXx patterns (Supplementary Fig. 3) in the above sense. In these other cases, the frequency of associated correlated flows is within the 10–20% range.

To check how the SF days contribute to the dynamical trend, we recalculated the dynamical trend excluding the SF days: we removed SF days from the time series, calculated the analogue temperatures of remaining days, the resulting yearly TXx, and recalculated the dynamical trend. We also did the opposite operation by keeping only SF days in the time series. On average over Western Europe (Fig. 2b), the dynamical TXx trend without SF patterns becomes insignificant over Western Europe (0.08 °C/GWD on average over Western Europe), while the SF-only TXx dynamical trend is both high and statistically significant (1.3 °C/GWD). Similar results are found when using a different reference date among the 10 most representative patterns. Dynamical TXx trends over Western Europe can therefore largely be explained by changes in the characteristics of SF patterns. First, their frequency has increased by 43% [10%;76%] per GWD (52% with time between 1950 and 2022) (see Supplementary Table 1). Second, the number of "events" (one event is defined here as a set of consecutive days) per year and their mean persistence have increased (see Supplementary Fig. 4). The persistence of SF patterns has increased by about 24% along the period [−1%, +50%] as a function o f GWD. Such changes all give more chance, within a season, to reach the high end of the conditional temperature distribution. Other characteristics may also have changed (eg. amplitude) but were not investigated here. Significant frequency increases are also found for at least the 10 most representative patterns of Supplementary Fig. 3, with rates in the range of 35% to 55%.

Note that SF is not the only flow pattern changing, and not all patterns associated with TXx days have an increasing frequency or persistence. For instance, the 23/07/2021 pattern, corresponding with summer TXx in central France for 2021, shows no particular evolution (Supplementary Fig. 4). Our results are also consistent with the increase in occurrence and persistence of the specific class of double jet circulations explaining a large fraction of European heat extremes[3], and about half (i.e., much more than the mean probability, 15%) of double-jet days are found within the SF days.

## Simulated temperature trends and their dynamical contributions

The representation of summer TXx and TXm trends has also been analyzed for a large number of CMIP6 model simulations (273 simulations in total for 36 models) (see Methods section for data processing). Over Western Europe, almost all CMIP6 simulations fail to simulate the observed strong TXx trends, as seen in Fig. 3a, plotting the percentage of simulations with larger trends than observed, for each grid point. These differences are less pronounced for TXm (Fig. 3b) but the number of runs reaching the ERA5 trend remains small here too (10-20% in large parts of South-Western Europe). There are also other land areas outside Western Europe where the CMIP6 simulations are mostly above the observed warming TXx trend (i.e. Sahara, Northern Scandinavia, Southern Balkans). This suggests that there is no general underestimation of extreme heat trends over all regions (or land regions). However, understanding these regional discrepancies across the globe is beyond the scope of this article.

When averaging TXx trends over the Western Europe region above defined, only 4 of the 273 individual runs analyzed (members of 3 models out of 36, ACCESS-ESM1, NorESM2-LM and KIOST-ESM) have a larger trend than the observations. The strong TXx trends observed correspond to the ~98-99th percentile of the overall CMIP6 distribution and could, from a statistical standpoint, be interpreted as consistent with Western Europe witnessing a very unlikely phase of low-frequency internal variability. However, in the five large model ensembles that were at our disposal (eg. ACCESS-ESM1-5, CanESM5, IPSL-CM6-LR, MIROC6, MPI-ESM1-2-LR), only ACCESS-ESM1-5 has a few members for which TXx warms as rapidly as observed (Fig. 3c), but this ensemble strongly overestimates the TXm trend (Fig. 3d). Hence, this ensemble does not correctly estimate the daily maximum temperature distribution as observed in ERA5.

Our results are qualitatively robust to the way trends are calculated. We estimated trends relative to time instead of GWD, and to each model initial-condition ensemble mean GWD instead of individual member GWD. In the first (resp. second) case, 9 (resp. 5) simulations (from 4 different models) slightly exceed the ERA5 TXx trend. Trends relative to time allowed in particular two members of CanESM5 to reach observations thanks to the strong global warming (about 1.7 °C since 1950), while the regional response to global warming (the regional trend as a function of GWD is about twice weaker than in ERA5.

We also implemented a multiple testing procedure, the False Discovery Rate[36-38], to test the significance of the result in Western Europe. Under the hypothesis that "models are indistinguishable from reality", the rank of the observed TXx and TXm trends in the distribution of members is uniform and there can be regions over which the observation falls outside the model range only by chance. Supplementary Fig. 5 shows that even taking into account the multiple nature of the test, Western Europe is among the regions where the mismatch between observed and simulated TXx trends is significant at the 95% confidence level in the sense of the FDR procedure, while no significant mismatch is found in this region for TXm trends.

Climate simulations do not capture the dynamical changes underlying these temperature extreme changes. We applied the analogue analysis to all available realizations for each model for which 500 hPa wind fields were available (170 simulations in total). This set was found to be rather representative of the overall simulation distributions, albeit with more weight on faster-warming simulations (see Fig. 3a, b histograms) regarding TXx trends. None of their dynamical TXx trends reach the amplitude of the observed one over Western Europe (Fig. 4a). This shows that there is less than 1% chance that the observed trend estimate is drawn from the same population as simulation estimates, accounting for all uncertainties. Remarkably, all members of the three available large ensembles (ACCESS-ESM1-5 [40 members], IPSL-CM6A-LR [31 members] and MPI-ESM1-LR [30 members]) exhibit values lower than observed, despite a few members exceeding the overall TXx trend. Also, on average over Western Europe, for TXm, a handful of models do have dynamical trends comparable to or larger than observations, but all others exhibit lower trends (Supplementary Fig. 6).

We also calculated the thermodynamical trend obtained as a residual by subtracting the dynamical trend from the total trend and reported the result in Fig. 4b. This shows that climate models exhibit thermodynamical contributions that are broadly consistent with ERA5, but there is a tendency for an underestimation of TXx thermodynamical trends, and a general agreement for TXm trends (see Supplementary Fig. 6). This analysis clearly shows that dynamical changes are largely responsible for the mismatch between modeled and observed TXx trends.

All 170 climate simulations realistically simulate the climatological mean frequency of the SF patterns (range from 12.5% to 18%).

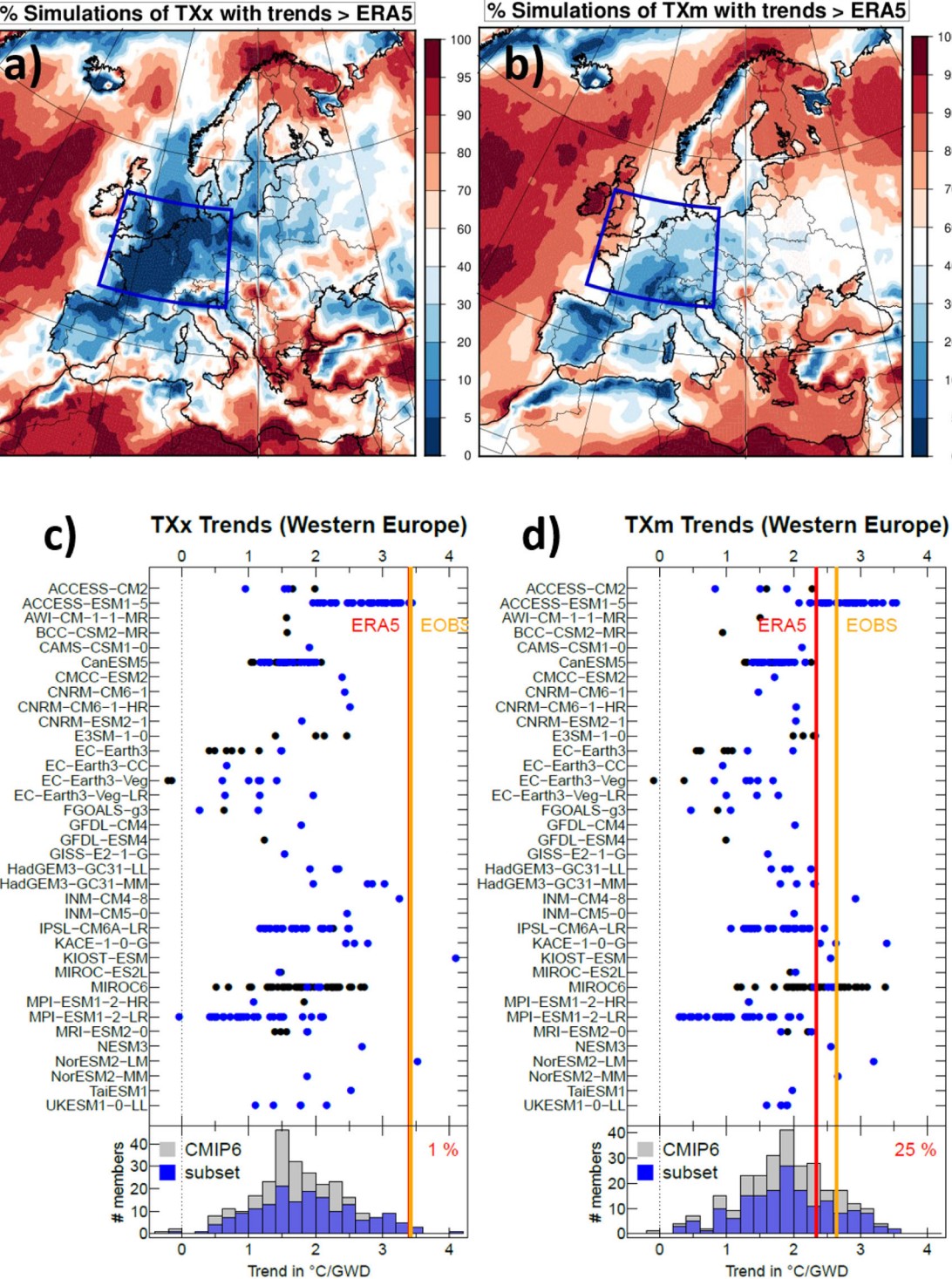

**Fig. 3 | Simulated vs. observed TX trends in Western Europe.** Comparison between the ECMWF reanalysis ERA5 and 273 CMIP6 simulations of trends in Summer maximum summer of daily maximum temperature, TX, (TXx, **a**) and **c**) and summer mean summer TX (TXm, **b**) and **d**) in °C/GWD represented in different ways; top panels: percentage of simulations with a trend larger than ERA5 at each grid point; bottom panels: representation of trends for model ensembles (dots) and observations (red and orange lines) after averaging over Western Europe (5°W to 15°E; 45°N-55°N); blue dots represent the 170 simulations that were analyzed with the analogue approach. Histograms at the bottom of the figure summarize the overall distribution of the TXx (left) and TXm (right) trends across the 273 simulations considered, together with the (blue) part analyzed with the analogue approach. Percentages of simulations with a trend larger than ERA5 are indicated in top right corners.

However, the rapid observed increase in frequency of this flow field (+43%/GWD [10−76%]) is only roughly captured by one among the 170 simulations (NorESM2-LM, and weaker in the others (Supplementary Table 1).

## Discussion

Overall, our results show that, except for a very few of them, CMIP6 simulations do not capture the rapid observed warming of extreme heat over Western Europe. The analysis of atmospheric

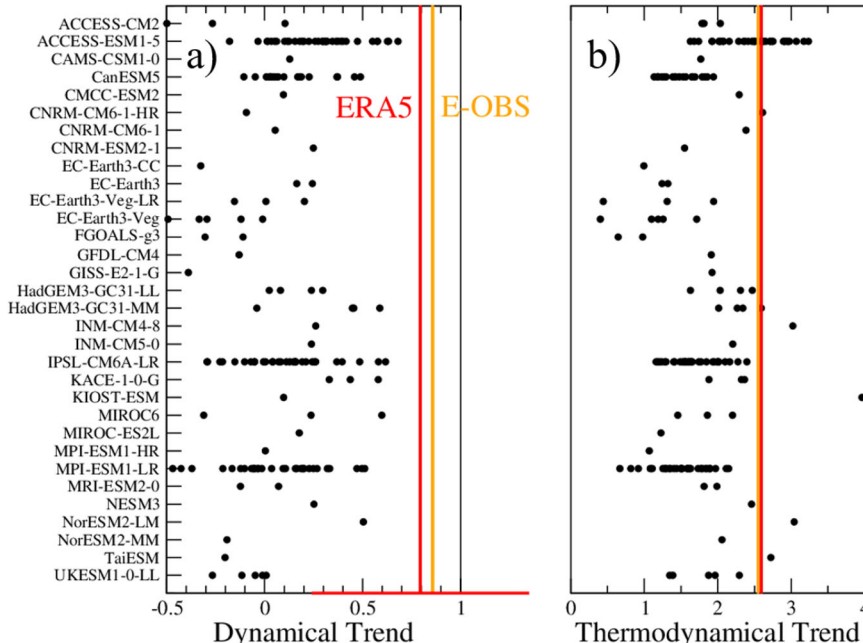

**Fig. 4 | Observed and dynamical and thermodynamical temperature trends.** Dynamical (**a**) and thermodynamical (**b**) contributions to the summer TXx (summer maximum of maximal daily temperature) trends from ERA5 ECMWF Reanalysis (red line), E-OBS observation (orange line), and the 170 CMIP6 model simulations (names in ordinate) that were available (black dots) averaged over Western Europe. The thermodynamical contributions are simply calculated as residual by subtracting the dynamical trend from the total trend (Fig. 3). For reference, the red bar at the bottom of **a** stands for the 95% confidence interval of the estimate of the circulation changes shows that there is a large dynamical contribution to this observed trend, which is underestimated in all the 170 climate simulations analyzed, explaining a large part of the discrepancy in trend between models and observations. By contrast, models and observational trends are broadly consistent in terms of the thermodynamic contribution to the trend in mean temperatures. Although it cannot be completely ruled out, the systematic mismatch between dynamical trends of 170 simulations and the observations, suggest that it is unlikely due to pure chance under the assumption of perfect models. We cannot either rule out other sources of systematic uncertainties such as lack of homogeneity of reanalyses, in particular for circulation patterns, or inaccuracies in the aerosol and land use forcing changes that would translate in systematic model/observation trend mismatches.

ERA5 TXx dynamical trend, estimated with a Gaussian assumption, i. e. the interval is calculated as plus or minus 2* the standard deviation (STD) of the error estimate on the trend coefficient. This confidence range describes the uncertainty related to the internal variability. This shows that this confidence range, calculated with the single realization of the observation, is consistent with the uncertainty range calculated from simulation members (respective standard deviations for observed trend and simulated trends of 0.28 and 0.25).

Determining the cause of model-observations dynamical trends mismatch is critical to assess whether the large observed warming TXx trend is likely or unlikely to continue. If due to a wrong forced dynamical regional response—models underestimate the forced response to greenhouse gases—then this mismatch is expected to remain and even strengthen in the future, as global warming increases. If related to unforced internal variability[39,40]—internal variability simulated by models is too small[41]—then the mismatch is expected to decrease in the future, but the term of this decrease is unknown and could be years or decades, leaving the fate of Western Europe heatwaves in large uncertainty.

Here we have shown that the observed extreme temperature trends for Western Europe are weaker in CMIP6 simulations than in observations, largely due to model dynamical trends systematically weaker than the observed ones. Similar conclusions were found for wintertime weather over Europe[42]. Note that there are also other regions on Earth where model TXx trends have large excursions from ERA5, but our study focused on Western Europe. Further research is needed to determine the causes of the mismatch between simulated and observed heat trends, whether this is due to uncaptured internal variability or missing (dynamical) forcing/processes. Either way, our results call for caution when using climate model projections for adaptation and resilience plans.

## Methods

### Calculation of dynamical contributions to mean and extreme summer temperature trends

The method used to estimate dynamical contribution to the change in one variable follows the conceptual framework developed in Vautard et al. (2016), with a different implementation here. It is based on the estimation of the change in the variable solely due to the changes in regional upper-air circulations. For instance, even without extra heating from radiative and diabatic processes, an increase in the frequency of southerly flows in Western Europe would induce a mean regional warming. An increase in anticyclonic conditions would similarly lead to increased radiation and thus temperature. This can also lead to a cooling if increasingly frequent circulations are linked to cooler temperatures (eg. in Northerly winds). To estimate this dynamical effect of changing circulations on temperatures, we need to carefully remove any thermodynamical effect of climate change.

We assume that daily temperature $T$ (which can be mean, minimum or maximum daily temperature, and in the current article will be maximum temperature) has a distribution at a given location or grid point which depends on the atmospheric circulation and on other processes, including global warming. We then assume a decomposition into:

$$T = \langle T|X \rangle_{GWD} + T' \tag{1}$$

where $X$ is the 500 hPa streamfunction anomaly, characterizing the atmospheric circulation (simultaneous to the temperature), $GWD$ stands for the global warming degree, $\langle T | X \rangle_{GWD}$ is the average daily maximum temperature conditioned to the circulation, assumed to be dependent on $GWD$, and $T'$ is a fluctuation. This circulation-conditioned temperature includes not only advection effects (i.e. from cooler/warmer regions), but also all processes linked to the circulation (subsidence in anticyclone, increased radiation, surface-atmosphere feedbacks, …), so the overall dynamical trend includes all underlying processes tied to the dynamical conditions. In order to remove thermodynamical effects due to climate change, we scale all temperatures to a reference warming level. For this, we assume that the circulation-conditioned mean temperature depends linearly on the global warming level, so the decomposition can be written:

$$T = \langle T | X \rangle_{ref} + b(X) \cdot \left( GWD - GWD_{ref} \right) + T' \qquad (2)$$

where *ref* refers to a reference global warming level, taken here as that of 2022, so all changes are expressed relative to 2022. The coefficient $b(X)$ represents the mean warming rate conditioned to the circulation $X$, which includes thermodynamical effects of the climate change response—it is therefore assumed that the amount of warming depends on the circulation type. Assuming one can calculate $b(X)$ and $GWD$, all daily temperatures are then scaled to the reference level with the following thermodynamical correction:

$$T_s = T - b(X) \cdot \left( GWD - GWD_{ref} \right) \qquad (3)$$

The dynamical contribution to any temperature trend constructed from daily temperatures (e.g. here TXm, TXx) can then be calculated from the $T_s$ time series, because changes with $GWD$ are only through the changes in the frequency of occurrences of $X$ for given GWDs. Trends should also not depend on the particular time $T_s$ values are drawn as long as they occur simultaneously to a streamfunction anomaly which is similar to that encountered in the same sequence order as that of the series. Hence to increase statistical robustness and remove any residual link to the specific order of temperatures, we replace $T_s$ temperatures by those occurring in circulations $X$ along the time series. This has the advantage of "randomizing" the timing of analogues and providing multiple realizations to calculate dynamical trends. A new temperature analogue series is created by replacing each daily with that of the best circulation analogue, then another new series is made with the second best analogue, etc… (see below for practical analogue calculation). From each of these analogue time series, TXm and TXx are recalculated for each year, then averaged across analogues, and a regression with GWD is calculated at each grid point, together with its confidence interval, (plus or minus twice the standard error of the regression coefficient). To keep analogue quality high, we limit the number of time series to 3. To calculate time series of averages over Western Europe land, we apply the 0.5°x0.5° land mask of E-OBS and average over the grid points included in [−5W − 15E; 45 N − 55 N].

### Estimation of yearly GWD
In practice, GWD is calculated as a moving centered 5-year average of the global temperature with available data, for reanalyses and models, accounting for series ends in ERA5 (i.e. for 1950, taking into account an average only over 1950 to 1952, and for 2022 an average over 2020 and 2021). The 2022 value is then subtracted to all values, so GWD is 0 in 2022, and generally negative before.

### Selection of circulation analogues
In practice, circulations are characterized by the 500 hPa streamfunction over the [−30 + 20°E; 30 60°N] domain. Analogs of a given

circulation are characterized by anomaly correlation coefficient (ACC) between streamfunction fields. For each summer day, we collect the best analogues (highest ACCs), and impose that they remain spaced by 6 days or more within a season, and self-analogues are not considered. This is done by successively testing fields in descending order of the ACC, and skipping days not respecting the separation with previously selected fields.

### Calculation of the circulation-conditioned thermodynamical trend *b(X)*
To calculate $b(X)$, we also use analogue circulations, in a different way than above: For each summer day $d$ of the 1950-2022 period, we estimate b(X($d$)) using a regression of each raw temperature T($d$) (before thermodynamical correction) associated with a large set of best analogue circulations of X($d$) found between 1950 and 2022 with the $GWD$ values of their respective year. We use the best 1% summer analogues (67 days) with the same spacing of at least 6 days. 99% of the worst of these 67 analogues across all summer days have ACC > 0.5, 65% have ACC > 0.7. Imposing a quality criterion on analogues such as ACC > 0.7 or more would leave days with an insufficient number of analogues for regression.

### Dynamical adjustment
Dynamical adjustment is used as a second, alternative technique to estimate the influence of circulation-induced temperature trends. This method relies on the idea that temperature variability can be decomposed into a component that is driven by circulation-induced variability, and a residual, thermodynamical component. The "thermodynamical" component is expect to contain a forced signal as well as any other unexplained variability or feedbacks[43]. Most applications of this technique characterize circulation-induced temperature variability using a proxy variable such as geopotential height[34,35,44,45]. Dynamical adjustment techniques typically rely on linear methods such as variants of linear regression or circulation analogue techniques.

Here, we use the spatial pattern of z500 in a relatively large circulation domain over Europe and the North Atlantic (−30 to 20°E, 30 to 60°N, similar to Fig. 1), following the method outlined in[46]. However, we introduce some modifications and additional details.We use a regularized regression technique, called "ridge regression", which is well-suited to deal with the large number of circulation predictor grid cells and a relatively short observed record. For TXx, we train our ridge regression model on the 15 warmest days in each summer during 1950-2021 at each grid cell in the ERA5 reanalysis, resulting in a total of 1080 observations (72 summers and 15 days per summer). Since the z500 field contains information about the lower troposphere, and is affected by temperature change via thermal expansion, we detrend the spatial z500 field by subtracting the global average z500 at each time step and over each grid cell in the circulation domain. Hence, the analysis is based only on relative changes within the z500 field. To obtain regional estimates of the circulation-induced component of TXx, we performed an area-weighted average across the grid cells within the study domain.

## Data availability
All analyzes have been conducted using 3 main data sets. The ERA5 reanalysis and the E-OBS data sets (processed from the https://climate.copernicus.eu) has been downloaded, and are available from the Climate Explorer https://climexp.knmi.nl. CMIP6 model simulations are available from the IPSL ESGF node https://esgf-node.ipsl.upmc.fr/.

## Code availability
Codes used in this article develop classical statistical algorithms, and are available upon request. Application codes are provided in the archive: https://zenodo.org/record/8310140.

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

## Acknowledgements

This study was partly supported by the European Union's Horizon 2020 research and innovation programme under grant agreement No 101003469 (XAIDA project). P.Y. was also supported by the grant ANR-20-CE01-0008-01 (SAMPRACE). The authors thank Dr. Efi Rousi for providing the sequences of dates of double-jet days. The authors also thank Atef Ben Nasser and the ESPRI IPSL data and computing service for their support in handling the large ensemble of climate simulations. The GMT v6.3 software is used for figure maps.

## Author contributions

R.V., J.C. and J.S. carried out the statistical analysis. T.H. provided the streamfunction fields for ERA5 and the calculation method. R.B., C.C., D.C., F.D., D.F., E.F., A.R., S.S. and P.Y. contributed to the design of the study and the interpretation of results. All authors contributed to the writing of the article.

## Competing interests

The authors declare no competing interest.
