## [Peer Review File · Nature Communications]

Heat extremes in Western Europe are increasing faster than simulated due to atmospheric circulation trendsReviewer #1 (Remarks to the Author):

This study examines trends in extreme heat over Western Europe in models and observations. The authors show that a substantial fraction of the observed trends in extreme heat are linked with changes in atmospheric circulation. The observed trends are on the extreme end of the model ensemble spread, but those few simulations that do capture the magnitude of the observed trends, do so for the wrong reasons. No simulation shows an increase in circulation-linked heat extremes as high as is seen in the models. The authors conclude that models are either missing a circulation response to external forcing or the models systematically underestimate internal variability – either would have important societal consequences.

Overall, I think that this is well done study on an important topic. The results are novel and, for the most part, convincing. This work will help motivate further research into the causes of the model-observations discrepancy. The paper is, in my opinion, potentially suitable for publication in the Nature Communications after the following comments addressed:

There are two issues that could point to authors potentially underestimating the chance that the model-observation difference is from internal variability that is accurately captured by the models.

-First, the authors are examining very small region that was not chosen a priori. It was chosen because the trends are very extreme. Even if models were perfect, if you looked at every small region there is a good chance that somewhere the trends will be very extreme due to internal variability, and potentially outside model spread. While the observed trends do look very extreme relative to the models at this location, the chances of trends this extreme occurring somewhere by chance are substantially higher. This is essentially the 'multiple hypothesis testing problem', where the null hypothesis is that the models and observations agree.

-The second issue is that the methods the authors use will likely artificially reduce the ensemble spread from internal in some models. As the authors are likely aware, a subset of the CMIP6 models have very high climate sensitivity and show too much global warming over the historical period relative to observations. By calculating the trends with respect to global mean surface temperature, the authors are essentially accounting for these differences in the ensemble means. However, the spread is artificially altered if the global mean trends are different to that of the observations. It is ratio of the global mean trends to the spread from internal variability that is maintained when calculating trends this way. If two models have the same spread due to internal variability in the raw trends with respect to time, but one model has twice as much global warming, when calculating the trends with respect global mean temperatures, this resulting model spread will be reduced by a factor of 2 relative to the other model. This is particularly an issue for the CanESM5 model which warms substantially more than observations and is one of the large ensembles used to conclude that the observed trends are outside the model spread. Not surprisingly, the ensemble spread in this model also appears to be quite small (Fig 3). Of course, models that warm less than observations will show too much spread, but I suspect that these are less common.

I don't think either of these are sufficient to completely invalidate the conclusions, but I think the authors should either account for these issues and/or make it clear that these issues exist.

Other comments/questions:

Figure 1: It would help if the different panels had the same colour scales.

L138-140: Do you have any insights into why the dynamical adjustment methods result in smaller trends? Although the methods are different, they are in principle trying to accomplish the same thing.

L216: It looks like UK-ESM1-0-LL only has 4 members (Fig 3). I wouldn't consider this a large ensemble.

L251-258/Figure 4: I am really surprised that most of the model spread in the trends in extremes comes from the thermodynamic component and that there is very little spread in the dynamical

component. We always hear about how the thermodynamic responses to climate change are better known and that the dynamics and associated impacts are much more uncertain (e.g. Shepherd 2014, 10.1038/ngeo2253). Any comment/speculation on why that is different here?

L295-298: It should be mentioned that there is evidence that climate models underestimate the multidecadal variability of the atmospheric circulation in the North Atlantic (see e.g. O'Reilly et al 2021, DOI:10.1038/s43247-021-00268-7, and references therein)

Reviewer #2 (Remarks to the Author):

Review of NCOMMS-23-02718-T: Heat extremes in Western Europe are increasing faster than simulated due to missed atmospheric circulation trends by R. Vautard et al.

Summary:

Summer temperatures and the amplitude of heat extremes in Western Europe have increased much faster than elsewhere in the mid-latitudes during the past decades, leading to frequent unprecedented heatwaves and threatening human lives. It is crucial to project how these events will change in the future so that mitigation and adaptation measures can be taken. The climate model is an essential tool to make such projections and possibly the only tool we can rely on. However, the models may not sufficiently capture the observed trends, and the decisions based on this biased projection may harm the adaptation and resilience plans. Based on the ERA5 and E-OBS datasets and 273 runs from the CMIP6, this study finds that a substantial fraction of the observed trends of the summer temperatures and heat extremes in Western Europe arises from the dynamical component due to changes in the atmospheric circulation, with a southerly flow pattern. It also suggests that the paces of the increasing magnitude of the summer temperatures and heat extremes in Western Europe are severely underestimated by the current state-of-the-art climate models. The reason is that the models underestimated or erroneously represented the southerly flow-like circulation response to external forcing. Moreover, it suggests that the few simulations reaching the observed warming trends in extreme heat are accompanied by a decrease in the occurrence of southerly flows, meaning that the models capture the warming trend for the wrong reason. These results are important because they imply a conservative projection and high uncertainties for future changes in the summer heat over Europe. They guarantee caution for stakeholders and decision-makers. The topic is relevant, the methods are reasonable, and the results are overall supported by the evidence provided. I recommend the authors revise their manuscript to further clarify several places. The following comments are for the authors' consideration.

Major comments:

1. The authors did not describe clearly what benchmark was used to calculate the analogue. It seems to be the 500hPa streamfunction on 29 June 2019 (line 147), but it might also be the 500hPa streamfunction on other dates or its trends (lines 106-108). Without an explicit description, I cannot judge the reliability of the subsequent analyses and results. Using the streamfunction on 29 June 2019 is OK. Still, the authors need to justify that the circulation in the 2019 heat extreme is typical enough to represent most of the heat extremes. If it is not typical, the subsequent analysis may be biased. I would also like the authors to discuss the sensitivity of the results to the choice of domain for the 500hPa streamfunction.
2. Following on from the previous comment, what is the most typical circulation configuration for the occurrence of heat extremes in Western Europe, the southerly flow pattern, or others? The analysis in this study indicates that the southerly flow pattern is important, but it does deny the importance of other patterns. There is a possibility that a second pattern is more dominant than the southerly flow pattern to cause heat extremes and that this second pattern is captured by models that reproduced the observed TXx trends. I strongly suggest the authors examine this possibility because it is directly relevant to the conclusion of this study.

Minor comments:

1. The authors interpreted that the modeled TXx trends are likely associated with internal interdecadal climate variability. I believe this is very likely the case. I can easily understand the

link between the trend of the TXm and the internal interdecadal climate variability, but I have no idea about the TXx case. Can the authors provide some explanations or interpretations?

2. Line 58. TXx and TXm-> TXm and TXx.

3. Line 122. What are the 'both cases'?

Reviewer #3 (Remarks to the Author):

This paper explores the mechanisms behind the strong positive trend of heat extremes in Western Europe over the past decades. The work separates the trend into dynamical and thermodynamical components, using two methods separate methods, showing that a dynamical trend has contributed around 25% of the total trend in annual summertime maximum temperatures (TXx). The research then goes on to explore this trend in the historical simulations of the CMIP6, finding that such a trend for this Western Europe region is produced in very few (3 out of 273 simulations, i.e. around 1%) of the CMIP6 simulations. The authors explore the mechanisms behind the trends in the CMIP6 simulations using the thermodynamic-dynamic separation, arguing that, even for models that exhibit a strong trend, the contribution from the dynamical component is too weak. They conclude that the strong trend in extreme heat seen over Western Europe is therefore either I. unforced internal variability, which is stronger than seen in the model simulations; or II. a forced trend due to anthropogenic warming that is missing in the models.

This work is original, provides interesting results, using recently developed methodologies to explore a topic of interest to the many people living in western Europe who have experienced these heat extremes in recent decades, as well as to others who are looking to this region as a possible sign of the potential impacts of climate change. The work is certainly of interest within the field of climate change, extreme events, and dynamical contributions; although I don't see any immediate significance to related fields.

My main concern is whether the findings are highly sensitive to the relatively small study region chosen. If natural (internal) variability does provide an unusually large contribution to the increasing trend, then, by design, the study has picked the one region that is more likely to be an outlier. This region has been studied precisely because there is a strong trend in extremes here, and it is important to understand whether this is a forced trend, or internal variability; however, I am not yet convinced by the presented results that the finding is as unexpected as the manuscript suggests.

The area of the study region ('Western Europe') is 20x10 degrees. There are 18x18 = 324 such areas on the globe. Given the strong degree of natural variability in the variables of interest, particularly TXx, some regions of the globe are likely to experience particularly anomalous trends, and I would expect at least one of these 324 regions is likely to experience a trend that is far into the tail of the distribution from the 273 CMIP6 simulations. Indeed, in Figs 3a,b, there is a lot of variability in this field (% of CMIP6 models with a trend greater than observed) with some regions of the Atlantic/northern Africa/Russia with over 90% or even over 95% of simulations exceeding the ERA5 TXm or TXx trend, and other regions with very low values, as mentioned in the text. I agree that understanding the global distribution of these values seems outside the scope of this article, but does this large variability not imply that perhaps 273 model simulations are not sufficient to sample internal variability for regions this small when looking at a highly variable metric such as TXx? The authors do mention this in line 214: 'could, from a statistical standpoint, be interpreted as consistent with Western Europe witnessing a very unlikely phase of internal interdecadal variability'; however, my feeling is that this is understated throughout the manuscript. I agree that it is very unlikely for any one specific region to experience this, but it is much less unlikely that one region somewhere on Earth is experiencing this internal variability, and that, by selecting the region based on the trend in TXx, you may have preferentially selected this exact region. I think this needs to be addressed throughout the manuscript. If you could show that no region of Earth saw as high a TXx trend as was seen over Western Europe, this would be a more convincing argument of a discrepancy between observations and the models.

That said, it is interesting that the models that do meet or exceed the observed trend do so for different reasons, and do not reproduce the observed dynamical trend. However, I again caution

about over-interpretation of a rare result in the context of looking only at one small region. Similar to my question above: does any model simulate the observed Western Europe dynamical trend for any region on Earth? I realise that we don't expect internal variability to be the same everywhere, but if you could show that the models don't simulate as strong a dynamical trend anywhere in the mid-latitude regions, that would be much more convincing that something unusual, and out of the range of the models, has been happening over Western Europe. I understand the logistical reasons for not running the circulation analogue analysis for all the ensemble members, but when you've picked one region out of 324 to study, using only 32 models simulations does not feel conclusive enough that the observations are so far outside what the models might suggest. Also, why not at least pick the ensemble member closest to the observed TXx trend for the ensemble simulations, since you're then comparing the dynamical trend to the observed dynamical trend? From Fig 4, I don't imagine this will make much difference, but it does seem a more reasonable approach.

I think the wording of the text also understates the possible role of internal variability (although it does acknowledge it in places). For example, line 246-247 states: 'Among those simulations with relatively similar total TXx trends as ERA5, none exhibit a statistically significant dynamical trend.' What about KIOST-ESM? This model exceeds the ERA5 trend in TXx, and the dynamical trend appears to be about half of the observed value – is this not large enough to be significant? How do you test the significance of the trends in the models and where is this shown?

Overall, I think this is an interesting study, and (with the exceptions of the points noted above) uses valid methodologies. To be published in Nature Communications, I think more concrete evidence (as suggested above) that the observations really are so far outside the modeled internal variability plus trend is needed.

Minor comments

You refer frequently to summer mean trends, when I think you mean summer mean daily maximum trends? Also 'warming trend' when you mean increase in TXm or TXx. This could be clearer – I think most readers will automatically think summer mean trends or 'warming trend' refers to mean temperature, not mean of maximum temperatures, even if you have defined it early in the paper.

For the analogue analysis carried out on the CMIP models, are the patterns you are looking for found in the models, or you find them in the observations.

Figures 1, 3 etc. You show TXm before TXx, but then discuss TXx first in the text – I suggest rewording to make your point more clear, and perhaps showing TXx first if that is your main point? It could also be more clear in the text that ERA5 is shown in Fig 1 and E-Obs is in S. Fig 1.

Line 63. Correction needed: Suppl. Fig. 1.

Line 75 'split mid-latitude jet states'

Line 82 'not robust and are sensitive to' (as written it's ambiguous whether they are, or are no sensitive to definition)

Line 108 'allows us'

Line 114. Could be re-phrased to be clearer, e.g. "we replace each daily temperature field by the temperature field from a different day that had an analogue circulation."

Line 119-121. This sounds like you don't trust that the method does actually remove all thermodynamic effects? Or is the lack of difference between the methods proof that it does? If so, could you rephrase as 'this was tested by re-calculating results with a scaling applied to all analogue temperatures....'.

Line 140. Is it correct to say that the 0.58C/GWD is equivalent to a linear change of 0.89C over 1950-2021? If so, this might be a better way of saying it. If not, please clarify what this linear change is, as you didn't report this for your analogues method.

Fig. 2b. A bit confusing to work out what the red and blue colours are – is it just that blue is for decreasing analogue trends and red is increasing analogue trends? This is a little confusing given that the decreasing one isn't significant (as inferred from the text). Is blue therefore 'not

significant' trends. I think a different color scheme could make this plot easier to interpret.

Line 165. What do you mean by 'the contribution of SF to yearly TXx becomes almost systematic'?

Line 193. Please specify what the 5% in parentheses represents? Is this the new (tripled) occurrence of double-jet days?

Extended Fig. 2. 'Circulation' is mis-spelled in the caption

Line 212. Extended data figure 3 doesn't seem like the correct figure to reference here.

Line 217, strongly overestimates

Line 217. The phrase 'warm as rapidly as observed' is not clear that you are referring to TXx, this sounds more like you're referring to mean temperatures. Suggest to rephrase for clarity.

Line 218 'overestimates the mean summer trend' do you mean average summer temperatures, which the phrase implies, but which you don't show, or are you referring to the average daily maximum temperatures as shown in Figure 3c? Please clarify.

Line 243. What is 'resp.' standing for in resp. maximum? Is it necessary?

REVIEWER COMMENTS

Reviewer #1 (Remarks to the Author):

This study examines trends in extreme heat over Western Europe in models and observations. The authors show that a substantial fraction of the observed trends in extreme heat are linked with changes in atmospheric circulation. The observed trends are on the extreme end of the model ensemble spread, but those few simulations that do capture the magnitude of the observed trends, do so for the wrong reasons. No simulation shows an increase in circulation-linked heat extremes as high as is seen in the models. The authors conclude that models are either missing a circulation response to external forcing or the models systematically underestimate internal variability – either would have important societal consequences.

Overall, I think that this is well done study on an important topic. The results are novel and, for the most part, convincing. This work will help motivate further research into the causes of the model-observations discrepancy. The paper is, in my opinion, potentially suitable for publication in the Nature Communications after the following comments addressed:

We thank the reviewer for the time spent and careful reading and comments. We hope to have addressed all comments. Please note that a few errors were found in the calculation of streamfunction: (the cdo calculation of vorticity requires a decreasing latitude grid, which was erroneously not what we used, and corrupted file problem for ERA5). The results are now replaced and the errors impacted essentially Figures 1 and Figure 4. In Figure 1 the dynamical trend is now about 20% smaller. In Figure 4, the mean summer temperature results (TXm) are significantly modified, with higher amplitude in TXm trends for some models. For TXx the results also show less difference between simulations and observations, but yet none of the analyzed simulation reaches the ERA5 dynamical trend amplitude. These changes do not alter the overall conclusions, and the abstract is reworded accordingly.

We then addressed all the questions raised by the reviewer, in particular the two major ones below.

There are two issues that could point to authors potentially underestimating the chance that the model-observation difference is from internal variability that is accurately captured by the models.

-First, the authors are examining very small region that was not chosen a priori. It was chosen because the trends are very extreme. Even if models were perfect, if you looked at every small region there is a good chance that somewhere the trends will be very extreme due to internal variability, and potentially outside model spread. While the observed trends do look very extreme relative to the models at this location, the chances of trends this extreme occurring somewhere by chance are substantially higher. This is essentially the 'multiple hypothesis testing problem', where the null hypothesis is that the models and observations agree.

The initial motivation to investigate this region was to focus on a region where regional attribution studies of extreme temperatures systematically exhibit a strong observed trend and a systematic underestimation in model trends. The region is highly populated, and we need to understand these trends for a proper adaptation. It was previously suggested that there are dynamical drivers causing this increase in Tx (e.g. double jet paper by Rousi et al. 2022), which motivated us to look further into this region as well.

That being said, based on comments from 2 of the reviewers on the choice of region and its relevance, we added two analyses.

The first one is an investigation of trends in TXx (averaging over land and sea) in each of the 2088 regions of the world of 20° longitude x 10° latitude between 75°S and 75°N shifted by steps of 5° longitude and 5° latitude (then allowing overlaps). Despite the initial region selection was not based on this fact, it turns out that the region selected has the highest trend of all the 2088 regions investigated. This is now mentioned in the discussion of Figure 1 : “ Interestingly, the $20^\circ \times 10^\circ$ Western Europe region has the highest TXx (all year round) trend of all the 2088 $20^\circ \times 10^\circ$ regions around the globe between 75°S and 75°N shifted by steps of 5° , even though the region was not adjusted for this purpose.”

The second one is the implementation of a multiple test: the False Discovery Rate (Benjamini & Hochberg, 1995; Wilks 2006, 2016), as suggested by the reviewer. The fraction of members with a trend higher than that of ERA5 behaves like a traditional p-value, since under H_0 “members are indistinguishable from reality”, its distribution is uniform over $[0, 1]$. For TXx trends, with $\alpha=0.1$ (equivalent to a 95% confidence level according to Wilks), we reject H_0 on several grid points, including some in our study region (see the Figure below which has been included as Extended Data Figure 5). This suggests that the model / obs discrepancy is significant even in the sense of a multiple testing procedure, and gives confidence that the result in Europe is not obtained by chance.

Wilks, D. S. (2006). On “field significance” and the false discovery rate. *Journal of applied meteorology and climatology*, 45(9), 1181-1189.

Wilks, D. (2016). “The stippling shows statistically significant grid points”: How research results are routinely overstated and overinterpreted, and what to do about it. *Bulletin of the American Meteorological Society*, 97(12), 2263-2273

Benjamini, Y., & Hochberg, Y. (1995). Controlling the false discovery rate: a practical and powerful approach to multiple testing. *Journal of the Royal statistical society: series B (Methodological)*, 57(1), 289-300.

-The second issue is that the methods the authors use will likely artificially reduce the ensemble spread from internal in some models. As the authors are likely aware, a subset of the CMIP6 models have very high climate sensitivity and show too much global warming over the historical period relative to observations. By calculating the trends with respect to global mean surface temperature, the authors are essentially accounting for these differences in the ensemble means. However, the spread is artificially altered if the global mean trends are different to that of the observations. It is ratio of the global mean trends to the spread from internal variability that is maintained when calculating trends this way. If two models have the same spread due to internal variability in the raw trends with respect to time, but one model has twice as much global warming, when calculating the trends with respect global mean temperatures, this resulting model spread will be reduced by a factor of 2 relative to the other model. This is particularly an issue for the CanESM5 model which warms substantially more than observations and is one of the large ensembles used to conclude that the observed trends are outside the model spread. Not surprisingly, the ensemble spread in this model also appears to be quite small (Fig 3). Of course, models that warm less than observations will show too much spread, but I suspect that these are less common.

On this point, the three ways of calculating trends and the argument in comment

I don't think either of these are sufficient to completely invalidate the conclusions, but I think the authors should either account for these issues and/or make it clear that these issues exist.

We understand the argument proposed by the reviewer but we think the different ways of calculating trends mean different things. The regional trend per degree of global warming (called GWD) is used here to represent the regional response to global warming whatever the global model warming rate is relative to time. As in many studies, and the IPCC report, regional consequences of climate change are studied per degree of warming (see eg. all figures of the WGI report). We agree then that, if the model spread does not vary with the climate sensitivity, its spread is reduced. But the word "artificially" is debatable. If a model has a too strong global warming with time and a correct regional TXx time trend, one may also expect strong regional responses and say that this good performance is artificial because of some error compensation.

In any case we have recalculated TXx trends relative to time and comparisons between model and observations provide qualitatively similar conclusions. In this case, 9(8) simulations have trends above ERA5(E-OBS) instead of 4(4). Now indeed 2 simulations from CanESM5 do have trends exceeding observations. However they have global temperatures that have warmed by about 1.7°C since 1950 while ERA5 witnessed a warming of 1°C. We also calculated trends in a third way, relative to the ensemble mean trend for each model, and found again only a small number of simulations 5(5) with trends exceeding observation. Our conclusions are therefore not qualitatively modified by the way trends are calculated. We added a paragraph:

"Our results are qualitatively robust to the way trends are calculated. We estimated trends relative to time instead of GWD, and to each model initial-condition ensemble mean GWD instead of individual member GWD. In the first (resp. second) case, 9 (resp. 5) simulations (from 4 different models) slightly exceed the ERA5 TXx trend. Trends relative to time allowed in particular two members of CanESM5 to reach observations thanks to the strong global warming (about 1.7°C since 1950), probably compensating another bias as the trends per GWD, assumed to represent the regional response to global warming are about twice weaker than in ERA5."

Other comments/questions:

Figure 1: It would help if the different panels had the same colour scales.

We did not change as the two trends (overall and dynamical) have very different upper bounds (+6°C/GWD and +1.5°C/GWD)

L138-140: Do you have any insights into why the dynamical adjustment methods result in smaller trends? Although the methods are different, they are in principle trying to accomplish the same thing.

This was at least partly due to the ERA5 streamfunction dataset, which was partly corrupted in the submitted version. Now the dynamical trend is smaller, 0.79°C/GWD, while the dynamical adjustment method, which now calculates gridpoint adjustment before averaging

over Western Europe, instead of the other way around, gives a dynamical trend of 0.56 °C/GWD. In the revised version, the dynamical trends from the dynamical adjustment and analogue approach are relatively closer than in the previous version. Dynamical adjustment shows relatively smaller trends, which may be because this approach is based on the principles of statistical learning, and trained regression models may not fully capture the circulation-induced variability. This may be caused by several factors, such as a limited training data or non-linear relationship between the predictors and target variables.

L216: It looks like UK-ESM1-0-LL only has 4 members (Fig 3). I wouldn't consider this a large ensemble.

The reviewer is right. The full sentence had a problem and was reformed.

L251-258/Figure 4: I am really surprised that most of the model spread in the trends in extremes comes from the thermodynamic component and that there is very little spread in the dynamical component. We always hear about how the thermodynamic responses to climate change are better known and that the dynamics and associated impacts are much more uncertain (e.g. Shepherd 2014, 10.1038/ngeo2253). Any comment/speculation on why that is different here?

This was due to the error in the streamfunction calculation which is now corrected (see above). Now the results show a larger variability in the dynamical trends, despite underestimated.

L295-298: It should be mentioned that there is evidence that climate models underestimate the multidecadal variability of the atmospheric circulation in the North Atlantic (see e.g. O'Reilly et al 2021, DOI:10.1038/s43247-021-00268-7, and references therein)

This study is now cited.

Reviewer #2 (Remarks to the Author):

Review of NCOMMS-23-02718-T: Heat extremes in Western Europe are increasing faster than simulated due to missed atmospheric circulation trends by R. Vautard et al.

Summary:

Summer temperatures and the amplitude of heat extremes in Western Europe have increased much faster than elsewhere in the mid-latitudes during the past decades, leading to frequent unprecedented heatwaves and threatening human lives. It is crucial to project how these events will change in the future so that mitigation and adaptation measures can be taken. The climate model is an essential tool to make such projections and possibly the only tool we can rely on. However, the models may not sufficiently capture the observed trends, and the decisions based on this biased projection may harm the adaptation and resilience plans. Based on the ERA5 and E-OBS datasets and 273 runs from the CMIP6, this study finds that a substantial fraction of the observed trends of the summer temperatures and heat extremes in Western Europe arises from the dynamical component due to changes in the atmospheric circulation, with a southerly flow pattern. It also suggests that the paces of the increasing magnitude of the summer temperatures and heat extremes

in Western Europe are severely underestimated by the current state-of-the-art climate models. The reason is that the models underestimated or erroneously represented the southerly flow-like circulation response to external forcing. Moreover, it suggests that the few simulations reaching the observed warming trends in extreme heat are accompanied by a decrease in the occurrence of southerly flows, meaning that the models capture the warming trend for the wrong reason. These results are important because they imply a conservative projection and high uncertainties for future changes in the summer heat over Europe. They guarantee caution for stakeholders and decision-makers. The topic is relevant, the methods are reasonable, and the results are overall supported by the evidence provided. I recommend the authors revise their manuscript to further clarify several places. The following comments are for the authors' consideration.

We thank the reviewer for the time spent and careful reading and comments. We hope to have addressed all comments. Please note that a few errors were found in the calculation of streamfunction: (the cdo calculation of vorticity requires a decreasing latitude grid, which was erroneously not what we used, and corrupted file problem for ERA5). The results are now replaced and the errors impacted essentially Figures 1 and Figure 4. In Figure 1 the dynamical trend is now about 20% smaller. In Figure 4, the mean summer temperature results (TXm) are significantly modified, with higher amplitude in TXm trends for some models. For TXx the results also show less difference between simulations and observations, but yet none of the analyzed simulation reaches the ERA5 dynamical trend amplitude. These changes do not alter the overall conclusions, and the abstract is reworded accordingly.

We then addressed all the questions raised by the reviewer, in particular the two major ones.

Major comments:

1. The authors did not describe clearly what benchmark was used to calculate the analogue. It seems to be the 500hPa streamfunction on 29 June 2019 (line 147), but it might also be the 500hPa streamfunction on other dates or its trends (lines 106-108). Without an explicit description, I cannot judge the reliability of the subsequent analyses and results. Using the streamfunction on 29 June 2019 is OK. Still, the authors need to justify that the circulation in the 2019 heat extreme is typical enough to represent most of the heat extremes. If it is not typical, the subsequent analysis may be biased. I would also like the authors to discuss the sensitivity of the results to the choice of domain for the 500hPa streamfunction.

We agree with the reviewer and now have increased the justification of the use of this date. We actually found that this date was the most representative of patterns occurring during TXx days in Central France, in the sense that it maximizes the correlation with all other TXx streamfunction patterns. This is now specified in the corresponding paragraph: "We select the reference date (29/06/2019) for which the streamfunction pattern (Fig. 2a) has a maximal average ACC (0.59) with other streamfunction patterns occurring each year when maximal temperature (TXx) is reached at this grid point, so it is most representative of those "TXx days"." In turn, we have also done all trends in occurrence for the 10 most representative patterns (in the same sense) and display the results in Extended Data Figure 3 and Extended Data Table 1, to show robustness of our findings. The corresponding paragraphs are modified accordingly in the text.

2. Following on from the previous comment, what is the most typical circulation configuration for the occurrence of heat extremes in Western Europe, the southerly flow pattern, or others? The analysis in this study indicates that the southerly flow pattern is important, but it does deny the importance of other patterns. There is a possibility that a second pattern is more dominant than the southerly flow pattern to cause heat extremes and that this second pattern is captured by models that reproduced the observed TXx trends. I strongly suggest the authors examine this possibility because it is directly relevant to the conclusion of this study.

As explained above, all trends in occurrence for the 10 most representative patterns (in the same sense) and display the results in Extended Data Figure 3 and Extended Data Table 1, to show robustness of our findings. It is important that the wording in the interpretation has changed since some of the patterns have frequency trends that are reproduced by one of the models, despite being generally underestimated by all others. This also alters one sentence in the abstract.

Minor comments:

1. The authors interpreted that the modeled TXx trends are likely associated with internal interdecadal climate variability. I believe this is very likely the case. I can easily understand the link between the trend of the TXm and the internal interdecadal climate variability, but I have no idea about the TXx case. Can the authors provide some explanations or interpretations?

We do not fully understand the question and to what part of the article or sentences the comment refers. We believe that both TXx and TXm trends are influenced by internal variability and climate change. Note that in the text wording, we now try to better balance between the two possibilities (internal variability and climate change response).

2. Line 58. TXx and TXm-> TXm and TXx.

Thanks, the typo is corrected

3. Line 122. What are the 'both cases'?

We now detail "with and without scaling". Note that the sentence has been rephrased for clarification.

Reviewer #3 (Remarks to the Author):

This paper explores the mechanisms behind the strong positive trend of heat extremes in Western Europe over the past decades. The work separates the trend into dynamical and thermodynamical components, using two methods separate methods, showing that a dynamical trend has contributed around 25% of the total trend in annual summertime maximum temperatures (TXx). The research then goes on to explore this trend in the historical simulations of the CMIP6, finding that such a trend for this Western Europe region is produced in very few (3 out of 273 simulations, i.e. around 1%) of the CMIP6 simulations. The authors explore the mechanisms behind the trends in the CMIP6 simulations using the thermodynamic-dynamic separation, arguing that, even for models that exhibit a strong

trend, the contribution from the dynamical component is too weak. They conclude that the strong trend in extreme heat seen over Western Europe is therefore either I. unforced internal variability, which is stronger than seen in the model simulations; or II. a forced trend due to anthropogenic warming that is missing in the models.

This work is original, provides interesting results, using recently developed methodologies to explore a topic of interest to the many people living in western Europe who have experienced these heat extremes in recent decades, as well as to others who are looking to this region as a possible sign of the potential impacts of climate change. The work is certainly of interest within the field of climate change, extreme events, and dynamical contributions; although I don't see any immediate significance to related fields.

We thank the reviewer for the time spent and careful reading and comments. We hope to have addressed all comments. Please note that a few errors were found in the calculation of streamfunction: (the cdo calculation of vorticity requires a decreasing latitude grid, which was erroneously not what we used, and a corrupted file problem for ERA5). The results are now replaced and the errors impacted essentially Figures 1 and Figure 4. In Figure 1 the dynamical trend is now about 20% smaller. In Figure 4, the mean summer temperature results (TXm) are significantly modified, with higher amplitude in TXm trends for some models. For TXx the results also show less difference between simulations and observations, but yet none of the analyzed simulation reaches the ERA5 dynamical trend amplitude. These changes do not alter the overall conclusions, and the abstract is reworded accordingly.

We then addressed all the questions raised by the reviewer, in particular the two major ones below.

My main concern is whether the findings are highly sensitive to the relatively small study region chosen. If natural (internal) variability does provide an unusually large contribution to the increasing trend, then, by design, the study has picked the one region that is more likely to be an outlier. This region has been studied precisely because there is a strong trend in extremes here, and it is important to understand whether this is a forced trend, or internal variability; however, I am not yet convinced by the presented results that the finding is as unexpected as the manuscript suggests.

The area of the study region ('Western Europe') is 20x10 degrees. There are $18 \times 18 = 324$ such areas on the globe. Given the strong degree of natural variability in the variables of interest, particularly TXx, some regions of the globe are likely to experience particularly anomalous trends, and I would expect at least one of these 324 regions is likely to experience a trend that is far into the tail of the distribution from the 273 CMIP6 simulations. Indeed, in Figs 3a,b, there is a lot of variability in this field (% of CMIP6 models with a trend greater than observed) with some regions of the Atlantic/northern Africa/Russia with over 90% or even over 95% of simulations exceeding the ERA5 TXm or TXx trend, and other regions with very low values, as mentioned in the text. I agree that understanding the global distribution of these values seems outside the scope of this article, but does this large variability not imply that perhaps 273 model simulations are not sufficient to sample internal variability for regions this small when looking at a highly variable metric such as TXx? The authors do mention this in line 214: 'could, from a statistical standpoint, be interpreted as

consistent with Western Europe witnessing a very unlikely phase of internal interdecadal variability'; however, my feeling is that this is understated throughout the manuscript. I agree that it is very unlikely for any one specific region to experience this, but it is much less unlikely that one region somewhere on Earth is experiencing this internal variability, and that, by selecting the region based on the trend in TXx, you may have preferentially selected this exact region. I think this needs to be addressed throughout the manuscript. If you could show that no region of Earth saw as high a TXx trend as was seen over Western Europe, this would be a more convincing argument of a discrepancy between observations and the models.

The initial motivation to investigate this region was to focus on a region where regional attribution studies of extreme temperatures systematically exhibit a strong observed trend and a systematic underestimation in model trends. The region is highly populated, and we need to understand these trends for a proper adaptation. It was previously suggested that there are dynamical drivers causing this increase in Tx (e.g. double jet paper by Rousi et al. 2022), which motivated us to look further into this region as well. Based on comments from 2 of the reviewers on the choice of region and its relevance, we added two analyses (see response to Reviewer #1).

That said, it is interesting that the models that do meet or exceed the observed trend do so for different reasons, and do not reproduce the observed dynamical trend. However, I again caution about over-interpretation of a rare result in the context of looking only at one small region. Similar to my question above: does any model simulate the observed Western Europe dynamical trend for any region on Earth?

I realise that we don't expect internal variability to be the same everywhere, but if you could show that the models don't simulate as strong a dynamical trend anywhere in the mid-latitude regions, that would be much more convincing that something unusual, and out of the range of the models, has been happening over Western Europe.

We did not investigate in detail the model / observation trend difference as this goes far beyond the scope of this article. However we do not expect models to systematically underestimate the dynamical trends worldwide, as different processes occur in different places. In particular the location of stationary waves, continents and mountain ranges may interact with the models dynamical biases. The suggestion of the reviewer to examine the dynamical trend patterns of models across the midlatitudes would however be very interesting for a global study that would go far beyond the purpose of this paper which is intended to have a regional focus. We also feel it would be difficult to interpret.

I understand the logistical reasons for not running the circulation analogue analysis for all the ensemble members, but when you've picked one region out of 324 to study, using only 32 models simulations does not feel conclusive enough that the observations are so far outside what the models might suggest. Also, why not at least pick the ensemble member closest to the observed TXx trend for the ensemble simulations, since you're then comparing the dynamical trend to the observed dynamical trend? From Fig 4, I don't imagine this will make much difference, but it does seem a more reasonable approach.

For sensitivity we have also used the member that has a highest TXx trend among those available for TXx and for the u and v fields (to calculate streamfunction) as suggested by the reviewer. Unfortunately this was only possible for 12 out of the 32 models, a reason for which we prefer to keep our selection based on the first available realization. Results are summarized in Extended Data Table 2 also reported below. While the TXx trends all increase (by construction), only 7 out of 12 dynamical trends increase, and never reach the ERA5 or EOBS dynamical trends.

We added : “For 12 of the models, ensembles we could have access to new ensemble members of ensembles that provided maximal TXx trend over Western Europe among available simulations. However these new members still exhibit underestimation in the dynamical trend and the SF frequency changes, and the dynamical trend was only increased in 7 cases, showing that the dynamical trend model underestimation does not (or weakly) depend on the member.”

Model_realization	TXx trend	TXx dynamical trend	Model_realization with largest TXx trend	TXx trend	TXx dynamical trend
ERA5	3.38	0.79			
EOBS	3.41	0.86			
ACCESS-CM2_r1i1p1f1	1.55	-0.27	ACCESS-CM2_r4i1p1f1	1.89	0.10
ACCESS-ESM1-5_r2i1p1f1	3.19	0.02	ACCESS-ESM1-5_r34i1p1f1	3.42	0.35
CanESM5_r1i1p1f1	1.89	0.23	CanESM5_r6i1p2f1	2.02	0.46
EC-Earth3_r1i1p1f1	1.49	0.16	EC-Earth3_r4i1p1f1	1.49	0.24
EC-Earth3-Veg_r1i1p1f1	1.14	-0.12	EC-Earth3-Veg_r4i1p1f1	1.42	-0.29
EC-Earth3-Veg-LR_r1i1p1f1	0.65	0.20	EC-Earth3-Veg-LR_r3i1p1f1	1.16	-0.15
FGOALS-g3_r1i1p1f1	0.26	-0.67	FGOALS-g3_r4i1p1f1	0.87	-0.11
HadGEM3-GC31-LL_r1i1p1f3	1.93	0.30	HadGEM3-GC31-LL_r4i1p1f3	2.49	0.02
KACE-1-0-G_r1i1p1f1	2.65	0.33	KACE-1-0-G_r3i1p1f1	2.80	0.44
MPI-ESM1-2-LR_r1i1p1f1	2.06	0.33	MPI-ESM1-2-LR_r3i1p1f1	2.09	0.20
MRI-ESM2-0_r1i1p1f1	1.87	-0.12	MRI-ESM2-0_r5i1p1f1	1.89	0.07
UKESM1-0-LL_r1i1p1f2	1.35	0.01	UKESM1-0-LL_r2i1p1f2	2.25	-0.05

I think the wording of the text also understates the possible role of internal variability (although it does acknowledge it in places). For example, line 246-247 states: ‘Among those simulations with relatively similar total TXx trends as ERA5, none exhibit a statistically significant dynamical trend.’ What about KIOST-ESM? This model exceeds the ERA5 trend in TXx, and the dynamical trend appears to be about half of the observed value – is this not large enough to be significant? How do you test the significance of the trends in the models and where is this shown?

Please note that we have now added the significance intervals in the dynamical trend estimates (Figure 4), and also that the results concerning this figure have changed due to streamfunction calculation error mentioned in the beginning. The paragraph has been completely reworded accordingly, and takes into account the reviewer’s remark. For instance we added “Confidence intervals however encompass the ERA5 and EOBS TXx trends, leaving the possibility that variability explains the mismatch.”.

Note also that Panels a and b of Figure 4 are now removed as most of the discussion and questions were on panels c and d and to save space because a significant amount of extra text has been added.

Overall, I think this is an interesting study, and (with the exceptions of the points noted above) uses valid methodologies. To be published in Nature Communications, I think more concrete evidence (as suggested above) that the observations really are so far outside the modeled internal variability plus trend is needed.

We hope to have responded to the reviewer's comments and questions. We have also tried to improve the text in several places.

Minor comments

You refer frequently to summer mean trends, when I think you mean summer mean daily maximum trends? Also 'warming trend' when you mean increase in TXm or TXx. This could be clearer – I think most readers will automatically think summer mean trends or 'warming trend' refers to mean temperature, not mean of maximum temperatures, even if you have defined it early in the paper.

We have now made sure that TXm or TXx are specified in all places where confusion could take place.

For the analogue analysis carried out on the CMIP models, are the patterns you are looking for found in the models, or you find them in the observations.

The analogue analysis is conducted in the same way over model simulations than over ERA5. The patterns are thus coming from the simulations, except when we investigate the SF frequencies, for which the observed pattern of ERA5 for 29/06/2019 is used.

Figures 1, 3 etc. You show TXm before TXx, but then discuss TXx first in the text – I suggest rewording to make your point more clear, and perhaps showing TXx first if that is your main point? It could also be more clear in the text that ERA5 is shown in Fig 1 and E-Obs is in S. Fig 1.

We followed the reviewer's suggestion and now systematically show the TXx results first. There is already a sentence mentioning ERA5 and E-OBS figure but the reference to Extended Data Figure 1 was not correct. This is now corrected.

Line 63. Correction needed: Suppl. Fig. 1.

Corrected

Line 75 'split mid-latitude jet states'

Corrected

Line 82 'not robust and are sensitive to ...' (as written it's ambiguous whether they are, or are no sensitive to definition)

Corrected

Line 108 'allows us'

Corrected

Line 114. Could be re-phrased to be clearer, e.g. "we replace each daily temperature field by the temperature field from a different day that had an analogue circulation."

We have followed this suggestion, but kept "the best" analogue.

Line 119-121. This sounds like you don't trust that the method does actually remove all thermodynamic effects? Or is the lack of difference between the methods proof that it does? If so, could you rephrase as 'this was tested by re-calculating results with a scaling applied to all analogue temperatures....'.

We have rephrased the sentence to better explain the temperature scaling: "As global warming is not homogeneous across the time period, and to ensure analogue regional temperatures represent a given global warming level to avoid we further apply a correction by scaling all analogue temperatures to a reference year for global warming (2022) (see Methods)."

Line 140. Is it correct to say that the 0.58C/GWD is equivalent to a linear change of 0.89C over 1950-2021? If so, this might be a better way of saying it. If not, please clarify what this linear change is, as you didn't report this for your analogues method.

The method has now been made more consistent with the analogues (trend averaged after grid-point calculation). New numbers are provided.

Fig. 2b. A bit confusing to work out what the red and blue colours are – is it just that blue is for decreasing analogue trends and red is increasing analogue trends? This is a little confusing given that the decreasing one isn't significant (as inferred from the text). Is blue therefore 'not significant' trends. I think a different color scheme could make this plot easier to interpret.

As explained in the caption, the red bullets are there for the annual TXx keeping SF flows only while the blue is for the annual TXx excluding the SF flows. We changed the color of the plain curve for the all-day TXx to avoid confusion and modified the legend. We hope this is more clear now.

Line 165. What do you mean by 'the contribution of SF to yearly TXx becomes almost systematic'?

We removed the confusing sentence as it did not bring important information.

Line 193. Please specify what the 5% in parentheses represents? Is this the new (tripled) occurrence of double-jet days?

This sentence was removed because the time periods of investigation are different and make it difficult to compare.

Extended Fig. 2. 'Circulation' is mis-spelled in the caption

Done

Line 212. Extended data figure 3 doesn't seem like the correct figure to reference here.

This has been corrected

Line 217, strongly overestimates

Corrected

Line 217. The phrase 'warm as rapidly as observed' is not clear that you are referring to TXx, this sounds more like you're referring to mean temperatures. Suggest to rephrase for clarity.

This has been rephrased to "However, in the five large model ensembles that were at our disposal (eg. ACCESS-ESM1-5, CanESM5, IPSL-CM6-LR, MIROC6, MPI-ESM1-2-LR), only ACCESS-ESM1-5 has a few members for which TXx warms as rapidly as observed, but this ensemble strongly overestimates the mean summer TXm trend (Figure 3dc). Hence, this ensemble does not correctly estimate the daily maximal temperature distribution as observed in ERA5."

Line 218 'overestimates the mean summer trend' do you mean average summer temperatures, which the phrase implies, but which you don't show, or are you referring to the average daily maximum temperatures as shown in Figure 3c? Please clarify.

Corrected, see above.

Line 243. What is 'resp.' standing for in resp. maximum? Is it necessary?

This paragraph has been almost completely rewritten.

Reviewer #1 (Remarks to the Author):

Overall, the authors have done a pretty good job addressing my comments. However, I still think the authors may be downplaying the chances that the observed trends are the result of internal variability that is correctly simulated by the models.

I like the inclusion of some results/discussion looking at field significance, but this is only for trends in TXx. The trend I am more interested in and concerned about is the 'dynamical contribution' trend (red dots in Fig 4). With the revised results, the observed trend is now closer to the modelled trends, and these results are based on fewer realizations. Given that the underestimation is confined to a small region in Western Europe, I don't see how you can rule out that the observed trend is a result of an unusual manifestation of internal variability. It may even be likely that a trend this unusual could appear somewhere on earth over a small region. From my understanding, it would not be feasible to repeat the field significance calculations for the dynamical contribution, so I would be satisfied with a short caveat about this.

L72: This point does not seem relevant. I can believe that the authors did not explicitly check this before doing the analysis, but I am sure the authors were aware that the trends in this region are very extreme and then chose their box to highlight the strong trends.

Reviewer #2 (Remarks to the Author):

Review of NCOMMS-23-02718A: Heat extremes in Western Europe are increasing faster than simulated due to missed atmospheric circulation trends by R. Vautard et al.

Recommendation: Accept

I thank the authors for addressing my previous comments reasonably and revising the manuscript accordingly. I am satisfied with the revision and believe it is a decent contribution to Nature Communications. Hence, I suggest accepting the manuscript in its present form.

Reviewer #3 (Remarks to the Author):

The revised manuscript is improved, particularly with the additional analysis, although corrections made to the analysis have somewhat weakened the strength of the results – the dynamical component is now smaller, and is more within the range of the model spread. My original concern remains, indeed, is perhaps stronger given the latest results, in that much of the text seems to imply that the observed trend is not caused by internal variability (whilst occasionally acknowledging that it could be), without showing much evidence for this. If the observations lie within the spread of the models, can we really say the models underestimate the trend? In particular, the title of the paper states that the models are missing the atmospheric circulation trends, but the latest results show that a number of ensemble members in fact do simulate the strength of the observed dynamical trend (although most of these simultaneously underestimate the observed thermodynamic trend, thus leading to an underestimation of the total trend). This suggests that the observed dynamical

trend in Western Europe is NOT outside of the range of modelled internal variability, and that there is substantial spread in the trends of circulation patterns over a 70 year period in the models, suggesting the potential for a strong role for internal variability. The observed trend is far above the mean modelled trend, but for un-biased models (which I accept is unlikely to be the case) we would expect roughly half the regions on Earth to show observed trends above the mean, and roughly half to show observed trends below the modelled mean (as demonstrated in the new Extended Fig. 5). Western Europe is high above the mean, but it does not stand out as particularly unusual in Extended Fig. 5; indeed, we expect Western Europe to be high above the modelled mean, as the region was chosen specifically because it has experienced a strong increase in heat extremes in recent years.

I understand the reasons behind the authors choice of region, and I'm not suggesting that they need to change the focus of the paper, but my opinion is that more emphasis needs to be made in the paper of the fact that they have, a priori, chosen a region that is likely to be in the higher (more unlikely, and maybe even highest, based on the new global analysis of TXx trends) range of modelled trends, and thus the observed values falling in the upper percentiles of the modelled range should not be surprising, and is not evidence that the models are missing the trends. I am not convinced that the presented analysis disproves the null hypothesis, that the TXx and TXm trends in Europe are a combination of model-captured trends, and internal-variability as diagnosed by large ensembles, and thus I still find some of the statements, including the title of the paper, mis-leading. I think there are some interesting points to be made with the analysis the authors have done, but I believe the focus chosen (models can't capture the observed trend) is over-simplified, particularly in the context of the latest, corrected, results. Given the importance of the topic of extreme weather, and the natural public interest, I believe it is critical that climate scientists are more careful to not under-state the role of natural variability in extreme weather events, particularly in papers that are likely to get noticed by the media, and I do not believe the science in this paper supports the statements that will be picked up from it (e.g. from the title). I think this is interesting research, and warrants publication, but only when sufficient care has been taken to not over-state the results, and under-state the possible role of natural variability.

Minor comments

Extended Fig. 3:

- This figure has a black background such that one cannot read the labels on the colorbar etc. Please change.

Line 85. Missing s in sensitive

Line 219-220: 'the Western Europe case is quite specific.' – this doesn't seem consistent with Extended Fig. 3, which shows that there are a number of regions around the globe where the observed trend is exceeded by most models, and a number of regions where the observed trend is not met by most models – this to me suggests that Western Europe may not be particularly unusual in terms of the ability of the models to reproduce the trends, further suggesting a strong role for natural variability? This should be emphasized more in this study. I agree that understanding the regional differences is beyond the scope of the study, but I do think they should be highlighted and acknowledged, and the implications for the conclusions of this study (i.e. either the models substantially overestimate trends in a number of regions AND substantially underestimate trends in

other regions, OR natural variability plays a substantial role.

Line 227. As I said in my previous review, I agree that it is unlikely that Western Europe would experience this phase of interdecadal variability, but it is much less unlikely that any region on Earth is experiencing it. The way to identify that particular region on Earth would be to select the one with the strongest observed trend, which is exactly what you have done here. Particularly given, in Extended data figure 3, Western Europe does not stand out as particularly unusual.

Line 256. Needs a space between Mean summer

Line 266. I'm not sure you can argue that the models underestimate the TXm trend at all? Given natural variability, if the observations lie within the spread of the models, can we say the models underestimate the trend? I also don't think you can say that the models don't capture the dynamical trend – 11 of the 32 models include the observed TXx trend within their 95% confidence interval, and indeed several models even show mean TXm trends of similar magnitude to the observations, so how is the conclusion that the models miss the observed trend?

Line 314. I would agree with your statements if you're referring to the mean of the CMIP6 models, which means that what is observed is unlikely to only be a forced trend, OR the models are missing a dynamical contribution to the trend. But, since the observed trends do fall within the model spread, I don't agree with the statement "CMIP6 simulations underestimate the rapid observed warming of extreme heat over Western Europe".

Line 333. Again, I would argue for "observed extreme temperature trends for Western Europe are not captured by the multi-model mean of CMIP6 simulations".

Reviewer #1 (Remarks to the Author):

Overall, the authors have done a pretty good job addressing my comments. However, I still think the authors may be downplaying the chances that the observed trends are the result of internal variability that is correctly simulated by the models.

I like the inclusion of some results/discussion looking at field significance, but this is only for trends in TXx. The trend I am more interested in and concerned about is the 'dynamical contribution' trend (red dots in Fig 4). With the revised results, the observed trend is now closer to the modelled trends, and these results are based on fewer realizations. Given that the underestimation is confined to a small region in Western Europe, I don't see how you can rule out that the observed trend is a result of an unusual manifestation of internal variability. It may even be likely that a trend this unusual could appear somewhere on earth over a small region. From my understanding, it would not be feasible to repeat the field significance calculations for the dynamical contribution, so I would be satisfied with a short caveat about this.

We thank again the reviewer for this remark regarding the dynamical trends. This incited us to strengthen our analysis by finally analyzing all simulations that were available on our servers or on other ESGF nodes. We ended up in calculating the TXx dynamical trends for 170 simulations. We think now the result is clear because none of them exceed 0.7°C/global warming degree while the ERA5 and E-OBS estimates are 0.8°C (see new Figure 4). In these sets of simulations we include 3 large ensembles clearly indicating that the dynamical contribution lies out of the distribution. Of course, an extraordinary situation where observation is outside of the 170-member ensemble cannot be ruled out but this is unlikely. This does not mean that the mismatch of the models is not an underestimation of the variability. The discussion is made at the end, and we tried to be prudent along the article, and reworded even the title not to induce potential interpretations.

L72: This point does not seem relevant. I can believe that the authors did not explicitly check this before doing the analysis, but I am sure the authors were aware that the trends in this region are very extreme and then chose their box to highlight the strong trends.

We removed the last part of this sentence.

Reviewer #2 (Remarks to the Author):

Review of NCOMMS-23-02718A: Heat extremes in Western Europe are increasing faster than simulated due to missed atmospheric circulation trends by R. Vautard et al.

Recommendation: Accept

I thank the authors for addressing my previous comments reasonably and revising the manuscript accordingly. I am satisfied with the revision and believe it is a decent contribution to Nature Communications. Hence, I suggest accepting the manuscript in its present form.

We thank the reviewer for all suggestions made

Reviewer #3 (Remarks to the Author):

The revised manuscript is improved, particularly with the additional analysis, although corrections made to the analysis have somewhat weakened the strength of the results – the dynamical component is now smaller, and is more within the range of the model spread. My original concern remains, indeed, is perhaps stronger given the latest results, in that much of the text seems to imply that the observed trend is not caused by internal variability (whilst occasionally acknowledging that it could be), without showing much evidence for this. If the observations lie within the spread of the models, can we really say the models underestimate the trend? In particular, the title of the paper states that the models are missing the atmospheric circulation trends, but the latest results show that a number of ensemble members in fact do simulate the strength of the observed dynamical trend (although most of these simultaneously underestimate the observed thermodynamic trend, thus leading to an underestimation of the total trend). This suggests that the observed dynamical trend in Western Europe is NOT outside of the range of modelled internal variability, and that there is substantial spread in the trends of circulation patterns over a 70 year period in the models, suggesting the potential for a strong role for internal variability.

We thank again the reviewer for the time spent in the second review and this critical remark. Actually, regarding dynamical trends, our previous Fig. 4a showed that none of the 32 models dynamical trends reached the observed one. Only some of the (roughly) estimated confidence intervals did. Still, only 32 simulations were considered to save computer burden. The reviewer remark incited us to strengthen our analysis by finally analyzing all simulations that were available (with all required fields) on our servers or on other ESGF nodes. We ended up in calculating the TXx dynamical trends for 170 simulations. We think now the result is clear because none of the 170 runs exhibit a dynamical trend as high as the one observed. The highest dynamical come close to 0.7°C/global warming degree while the ERA5 and E-OBS estimates are 0.8°C (see new Figure 4). In these sets of simulations we include 3 large ensembles. They clearly indicate that the observed dynamical contribution lies out of the model distribution. Given the large number of simulations now considered, the range of variability is essentially covered, and observations are clearly above all simulations. The fact that not even one simulation over 170 can catch the observed value is a very strong result in our view.

Of course, this still does not preclude the possibility that the model distribution is correct, and the observed dynamical trend is an extraordinary realization. But this seems very unlikely, and a standard statistical test would reject such a null-hypothesis. We added a sentence to mention this unlikely possibility in the discussion section : “Although it cannot be completely ruled out, the systematic mismatch between dynamical trends of 170 simulations and the observations, together with the field significance testing, suggest that it is unlikely that such mismatch is due by chance.”

Still, this mismatch could be due to an underestimation of the variability in models, and that is mentioned in the article. Given the recommendation of the reviewer, we tried to be prudent along the article, and reworded even the title not to induce potentially wrong interpretations. We hope this responds to the reviewer comment.

The observed trend is far above the mean modelled trend, but for un-biased models (which I accept is unlikely to be the case) we would expect roughly half the regions on Earth to show observed trends above the mean, and roughly half to show observed trends below the modelled mean (as demonstrated in the new Extended Fig. 5). Western Europe is high above the mean, but it does not stand out as particularly unusual in Extended Fig. 5; indeed, we expect Western Europe to be high above the modelled mean, as the region was chosen specifically because it has experienced a strong increase in heat extremes in recent years.

I understand the reasons behind the authors choice of region, and I'm not suggesting that they need to change the focus of the paper, but my opinion is that more emphasis needs to be made in the paper of the fact that they have, a priori, chosen a region that is likely to be in the higher (more unlikely, and maybe even highest, based on the new global analysis of TXx trends) range of modelled trends, and thus the observed values falling in the upper percentiles of the modelled range should not be surprising, and is not evidence that the models are missing the trends. I am not convinced that the presented analysis disproves the null hypothesis, that the TXx and TXm trends in Europe are a combination of model-captured trends, and internal-variability as diagnosed by large ensembles, and thus I still find some of the statements, including the title of the paper, mis-leading. I think there are some interesting points to be made with the analysis the authors have done, but I believe the focus chosen (models can't capture the observed trend) is over-simplified, particularly in the context of the latest, corrected, results. Given the importance of the topic of extreme weather, and the natural public interest, I believe it is critical that climate scientists are more careful to not under-state the role of natural variability in extreme weather events, particularly in papers that are likely to get noticed by the media, and I do not believe the science in this paper supports the statements that will be picked up from it (e.g. from the title). I think this is interesting research, and warrants publication, but only when sufficient care has been taken to not over-state the results, and under-state the possible role of natural variability.

We agree that based on Figure 3 only, strong wording could not be made on model/observation mismatch. However the new Figure 4 shows that it is unlikely that the systematic dynamical trends mismatch is due to pure chance (all 170 models are below observations). This figure shows that, in terms of the dynamical contribution to the trend, observations lie out of the model ensemble(s) distribution. We however agree that (as usual) this possibility cannot completely be ruled out. We also agree that the extreme heat trends in Western Europe are probably a combination of several factors, including variability, dynamical and thermodynamical responses, and we hope now the improved wording (including title) removes potential misinterpretations.

Minor comments

Extended Fig. 3:

- This figure has a black background such that one cannot read the labels on the colorbar etc. Please change.

The problem has been solved

Line 85. Missing s in sensitive

Correction has been made

Line 219-220: 'the Western Europe case is quite specific.' – this doesn't seem consistent with Extended Fig. 3, which shows that there are a number of regions around the globe where the observed trend is exceeded by most models, and a number of regions where the observed trend is not met by most models – this to me suggests that Western Europe may not be particularly unusual in terms of the ability of the models to reproduce the trends, further suggesting a strong role for natural variability? This should be emphasized more in this study. I agree that understanding the regional differences is beyond the scope of the study, but I do think they should be highlighted and acknowledged, and the implications for the conclusions of this study (i.e. either the models substantially overestimate trends in a number of regions AND substantially underestimate trends in other regions, OR natural variability plays a substantial role).

We removed “, and that the Western Europe case is quite specific”. We agree that model-obs inconsistencies are found in several regions (either model overestimation or model underestimation), but the TXx trend mismatch over Western Europe was found to be robust even considering this other regions (see the FDR test result).

Our new results conclude that systematic dynamical trends mismatch (for all 170 simulations) are unlikely due to chance under the assumptions of perfect models, and that this explains a large part of the TXx trend mismatch, which does not exclude variability as another factor. The fact that it could play a role in explaining the dynamical trend mismatch for Western Europe was already mentioned in lines 328-329 and 336-337, but not mentioned for other regions. We added a sentence for this in order to show that this is not a systematic result and reworded (more neutral terms). There were also some confusions between the dynamical trends and the TXx trends mismatch, which are, we hope, clarified now.

Line 227. As I said in my previous review, I agree that it is unlikely that Western Europe would experience this phase of interdecadal variability, but it is much less unlikely that any region on Earth is experiencing it. The way to identify that particular region on Earth would be to select the one with the strongest observed trend, which is exactly what you have done here. Particularly given, in Extended data figure 3, Western Europe does not stand out as particularly unusual.

We agree that Extended Data Figure 5 is a statistical field confidence test not visually making Europe stand out. However we try along this study not to stick to the statistical analysis of temperatures but try to explain the model/observation differences by disentangling dynamical and thermodynamical contributions. The stronger new result is in the origin of the mismatch, and the systematic model/obs dynamical trend mismatch over Europe, which may not be the same reason in other areas.

Line 256. Needs a space between Mean summer

Done

Line 266. I'm not sure you can argue that the models underestimate the TXm trend at all? Given natural variability, if the observations lie within the spread of the models, can we say the models underestimate the trend? I also don't think you can say that the models don't capture the dynamical trend – 11 of the 32 models include the observed TXx trend within their 95% confidence interval, and indeed several models even show mean TXm trends of similar magnitude to the observations, so how is the conclusion that the models miss the observed trend?

We agree about the sentence in Line 266, and removed “In addition to the underestimate trends,”. In the new analysis we have many more simulations and ensembles, showing a better picture of the uncertainty due to variability.

Line 314. I would agree with your statements if you're referring to the mean of the CMIP6 models, which means that what is observed is unlikely to only be a forced trend, OR the models are missing a dynamical contribution to the trend. But, since the observed trends do fall within the model spread, I don't agree with the statement “CMIP6 simulations underestimate the rapid observed warming of extreme heat over Western Europe”.

We reworded to: “Overall, our results show that, except for a very few of them, the CMIP6 simulations do not capture the rapid observed warming of extreme heat over Western Europe.”

Line 333. Again, I would argue for “observed extreme temperature trends for Western Europe are not captured by the multi-model mean of CMIP6 simulations”.

This one is more general, and we modified it to: “Here we have shown that the observed extreme temperature trends for Western Europe are weaker in CMIP6 simulations than in observations, largely due to model dynamical trends systematically weaker than the observed ones”

Reviewer #3 (Remarks to the Author):

The additional analysis with more models and ensemble members strengthens the conclusions of the paper, and, with the added text to highlight the possibility that models may be underestimating variability, and/or Europe is experiencing a particularly unusual mode of natural variability, and the corresponding change to the paper title, the authors have satisfactorily addressed my previous concerns. I have only one major concern remaining based on this latest submission.

Major concern:

I'm concerned by the fact that the confidence intervals on the trends (including dynamical contribution) are not shown in the figure in this most recent submission, or discussed in the text in the context of the model results. In the previous Fig. 4, the 95% confidence interval for the TXx ERA5 dynamical trend reached from 0.24 (a value that IS captured by many ensemble members) to 1.35 (far outside any model). In the latest Fig. 4, there are no uncertainty/confidence intervals shown. Trends in values associated with extremes can often be quite noisy (as perhaps evidenced by the large confidence range in the old version), and so I think consideration of the uncertainty in the ERA5 calculated trend is a critical part of this analysis, given the main message of the paper – that models do not capture the ERA5 value. In the response to reviewers, the authors mention that the uncertainty intervals were only roughly estimated – I think a more precise estimate of the uncertainty of the ERA5/EOBS trend, and more discussion of this in the context of the model results, is critical for accurate interpretation of your results. I would agree that, with the large number of model simulations now, error estimates on each ensemble member are not necessary, but I think a confidence interval of your observed trend value is critical – given the uncertainty in the ERA5 trend, how confident should we be that the value does fall outside of all of the model estimates?

Minor suggestions:

I still think that Fig. 3a and 3b contain some interesting information about the models and variability – whilst around 0 models show TXx trends > ERA5 for parts of Western Europe, in other regions, 95-100% of models have TXx trends exceeding the observed trend. I understand that the focus is on Western Europe, and the authors do mention that other regions 'have large excursions', but I think the authors could use this extra information to inform their discussion about the likeliness of different explanations for the discrepancy between models – indeed, I would suggest that this figure supports their suggestion that variability in the models may be underestimated (although the large scale pattern in Fig. 3a could of course be caused by a missing dynamical trend, i.e. a shift in wave phase, that is not captured by the models)

In Fig. 4, the left panel has some spurious black squares in the corners, and the number '33' in the bottom left, that the authors may wish to tidy up before publication.

Reviewer #3 (Remarks to the Author):

The additional analysis with more models and ensemble members strengthens the conclusions of the paper, and, with the added text to highlight the possibility that models may be underestimating variability, and/or Europe is experiencing a particularly unusual mode of natural variability, and the corresponding change to the paper title, the authors have satisfactorily addressed my previous concerns. I have only one major concern remaining based on this latest submission.

We thank again the reviewer for the careful consideration of our revision and added material. Our answers are below and we hope to have brought all explanations.

Major concern:

I'm concerned by the fact that the confidence intervals on the trends (including dynamical contribution) are not shown in the figure in this most recent submission, or discussed in the text in the context of the model results. In the previous Fig. 4, the 95% confidence interval for the TXx ERA5 dynamical trend reached from 0.24 (a value that IS captured by many ensemble members) to 1.35 (far outside any model). In the latest Fig. 4, there are no uncertainty/confidence intervals shown. Trends in values associated with extremes can often be quite noisy (as perhaps evidenced by the large confidence range in the old version), and so I think consideration of the uncertainty in the ERA5 calculated trend is a critical part of this analysis, given the main message of the paper – that models do not capture the ERA5 value. In the response to reviewers, the authors mention that the uncertainty intervals were only roughly estimated – I think a more precise estimate of the uncertainty of the ERA5/EOBS trend, and more discussion of this in the context of the model results, is critical for accurate interpretation of your results. I would agree that, with the large number of model simulations now, error estimates on each ensemble member are not necessary, but I think a confidence interval of your observed trend value is critical – given the uncertainty in the ERA5 trend, how confident should we be that the value does fall outside of all of the model estimates?

We have added the 95% confidence interval on the observation trend, estimated with a Gaussian assumption, i. e. the interval is calculated as plus or minus 2σ the standard deviation (STD) of the error estimate on the trend coefficient. This confidence range describes the uncertainty related to the internal variability, and can be interpreted as a confidence range of the forced response trend. The intersection between this interval and the interval encompassing all model estimates is not empty. However we think it can be misleading to interpret the probability that the “true” observed forced response trend falls within the interval of model simulated trends using both intervals and this overlap. Indeed, each single simulation trend estimate includes the same uncertainties as the observation estimates as they are calculated exactly in the same manner (in particular, it includes one single realization of internal variability). Comparing the observation confidence interval with the highest simulation value would mean double counting of the uncertainty related to internal variability.

Simulation trend estimates can be considered as random drawings from a distribution (and not an interval), which describes the uncertainty due to internal variability and model uncertainty. The question is therefore whether the observed trend estimate, calculated in the

same manner as for simulations, can be another drawing from the same population, and to reject this possibility with a certain probability of being wrong. Here, given that the observation estimate exceeds the 170 model estimates, the chance that the best estimate of observation trend is drawn from the same population as that of the simulation estimates, accounting for all kinds of uncertainties, is less than 1%.

We added in the text commenting Figure 4 : “This shows that there is less than 1% chance that the observed trend estimate is drawn from the same population as simulation estimates, accounting for all uncertainties”.

We also added in the Figure 4 caption: “For reference, the red bar at the bottom of Figure 4a stands, for the 95% confidence interval of the estimate of the ERA5 TXx dynamical trend, estimated with a Gaussian assumption, i. e. the interval is calculated as plus or minus 2* the standard deviation (STD) of the error estimate on the trend coefficient. This confidence range describes the uncertainty related to the internal variability. This shows that this confidence range, calculated with the single realization of the observation, is consistent with the uncertainty range calculated from simulation members (respective standard deviations for observed trend and simulated trends of 0.28 and 0.25).”

Minor suggestions:

I still think that Fig. 3a and 3b contain some interesting information about the models and variability – whilst around 0 models show TXx trends > ERA5 for parts of Western Europe, in other regions, 95-100% of models have TXx trends exceeding the observed trend. I understand that the focus is on Western Europe, and the authors do mention that other regions ‘have large excursions’, but I think the authors could use this extra information to inform their discussion about the likeliness of different explanations for the discrepancy between models – indeed, I would suggest that this figure supports their suggestion that variability in the models may be underestimated (although the large scale pattern in Fig. 3a could of course be caused by a missing dynamical trend, i.e. a shift in wave phase, that is not captured by the models)

We are not sure we can conclude from the figure that variability in models could be underestimated, because the large excursions can either arise from lack of variability but also from displacement of stationary waves or missing frequency of weather patterns. We prefer to not favor one explanation over the other.

In Fig. 4, the left panel has some spurious black squares in the corners, and the number ‘33’ in the bottom left, that the authors may wish to tidy up before publication.

This will be fixed in the final ps version of the figure which will be transmitted to the technical editor.